biomaterials/mechanical engineering/biomechanics

skin collagen fibre, stress relaxation, Poisson's ratio, digital image correlation, auxeticity, micro-CT

**Authors for correspondence:**
Sachin Kumar
e-mail: sachin@iitrpr.ac.in
Navin Kumar
e-mail: nkumar@iitrpr.ac.in

†These authors contributed equally to this study.

# Effect of collagen fibre orientation on the Poisson's ratio and stress relaxation of skin: an *ex vivo* and *in vivo* study

Krashn Kumar Dwivedi[1,†], Piyush Lakhani[2,†], Sachin Kumar[2] and Navin Kumar[1,2]

[1]Department of Biomedical Engineering, and [2]Department of Mechanical Engineering, Indian Institute of Technology, Ropar, India

 NK, 0000-0002-7958-8155

During surgical treatment skin undergoes extensive deformation, hence it must be able to withstand large mechanical stresses without damage. Therefore, understanding the mechanical properties of skin becomes important. A detailed investigation on the relationship between the three-dimensional deformation response of skin and its microstructure is conducted in the current study. This study also discloses the underlying science of skin viscoelasticity. Deformation response of skin is captured using digital image correlation, whereas micro-CT, scanning electron microscopy and atomic force microscopy are used for microstructure analysis. Skin shows a large lateral contraction and expansion (auxeticity) when stretched parallel and perpendicular to the skin tension lines, respectively. Large lateral contraction is a result of fluid exudation from the tissue, while large rotation of the stiff collagen fibres in the loading direction explains the skin auxeticity. During stress relaxation, lateral contraction and fluid effluxion from skin reveal that tissue volume loss is the intrinsic science of skin viscoelasticity. Furthermore, the results obtained from *in vivo* study on human skin show the relevance of the *ex vivo* study to physiological conditions and stretching of the skin during its treatments.

## 1. Introduction

Biomechanical performance of skin is necessary to maintain its primary functions, i.e. protection of underlying tissue, sensory regulation via fluids and heat exchange, homeostasis, etc. Skin is

the outermost layer of the body and shares about 15% of the whole-body weight, and has a surface area of approximately 1.5–2.0 m$^2$ [1]. Its structure is a composite of three distinct layers: the epidermis (outer layer), the dermis (middle and the thickest layer) and the hypodermis (innermost layer). Out of these three layers, the dermis is the main layer that provides mechanical strength to the skin. It is composed of elastin (approx. 2–4% of the dry skin weight), collagen fibres (approx. 70% of the dry skin weight, and are main load-bearing components) and ground substance (proteoglycans (PGs), water, etc.) [2–6]. The dermis holds about 60–70% of total skin water, which is entrapped by glycosaminoglycans (GAGs) macromolecules [7] and provides time-dependent mechanical behaviour to the tissue [8,9]. In addition to this, water mobility also determines the tissue ability to adapt to deformation through contraction and expansion of the collagen fibre network [10].

Several experimental techniques have been applied to characterize the skin mechanical properties, involving uniaxial tension [11–20], suction [21,22] and bulging [23–25]. The outcome of these studies shows an anisotropic, nonlinear elastic and viscoelastic behaviour of the skin. To depict the anisotropic behaviour of skin, Langer proposed lines along the direction of maximum tension, and these lines are called Langer lines or skin tension lines (STLs) [26]. The anisotropy of the skin is related to the organization of collagen fibres [25]. On the other hand, the viscoelastic behaviour of the skin originates through the interaction among its different constituents, such as collagen fibre, PGs and water [27]. Viscoelastic property of soft tissues plays a vital role in protective function (through dissipation of applied load) [15], pain sensation (by decreasing the discharge frequency of skin nociceptor as a result of decay in stress due to viscous dissipation) [28], drug delivery (the value of force required to puncture the skin depends on the penetration rate) [29], tissue expansion/plastic surgery (tissue expansion rate and the amount may depend on the creep rate of the tissue) [30] and wound healing (short stress relaxation improves the wounding quality, i.e. scar-less wound formation) [31].

The viscoelasticity of skin and other soft tissues is typically characterized through *ex vivo* tests, i.e. stress relaxation, creep and dynamic mechanical analysis [11,32–35]. Out of these techniques, stress relaxation is commonly used to investigate the viscoelastic behaviour of tissue as it is experimentally simple and can capture the long-term viscoelastic response of the tissue [11,14,18,36,37]. The stress relaxation phenomenon in soft tissues describes the time-dependent decay of mechanical stresses in response to step strain. Liu & Yeung [36] found that the amount of stress decay in the skin depends on the applied value of strain and specimen orientation; similar results were also observed for the pig skin [11]. Despite these studies, the underlying mechanism of stress relaxation in the skin is still undetermined. Albeit, the knowledge of these mechanisms is essential for the homeostasis/mechanotransduction of tissue as well as to understand the association of distinct constituents of skin with this mechanical behaviour. From a clinical point of view, stress relaxation can be helpful for the large wound closure, where a short stress relaxation will reduce the excessive stresses on the skin in the vicinity of the sutures as well as on the suture itself. This technique will reduce the tear of the skin, failure of the suture and the chance of necrosis [31,38,39].

In addition to this, in most of the studies, the skin and other soft tissues are treated as incompressible materials [14–16,34,40]. The assumption of incompressibility for hydrated materials is only justified if the interstitial fluid (IF) is bound to the tissue matrix or the specimen is entirely confined during loading. Brown *et al.* [41] demonstrated that under unconfined loading, the sclera tissue behaves like poroelastic material; a similar result was also observed by Wang *et al.* [42] for the skin tissue. Some other studies have reported the value of Poisson's ratio for skin [11,43,44], ligament and tendon [45–47] beyond the limit of incompressible material (−1 to 0.5). Besides these studies, a detailed description of the underlying science of this mechanical response of the skin is missing in literature; also, the cause of stress relaxation in the skin is not explained till now. This motivates us to investigate the intrinsic science of compressible and stress relaxation behaviour of the skin, which will help in the treatment of skin disorders.

The work presented in this article aims to understand more insight into the mechanical behaviour of skin, from the uniaxial tests performed on the pig (*ex vivo*) and left-back region of human skin (*in vivo*). In the present study, pig skin is chosen because of the similarities with human skin [11,48]. In this study, the digital image correlation (DIC) technique is coupled with a uniaxial tensile test machine to capture the three-dimensional deformation response of pig skin, whereas an in-house build tensile test set-up is coupled with a digital microscope to capture the *in vivo* deformation response of human skin. Micro-CT, scanning electron microscope (SEM) and atomic force microscope (AFM) are used to connect the change in collagen arrangement with three-dimensional deformation (change in Poisson's ratio) response and stress relaxation behaviour of the skin. Further, for the first time, this study demonstrates the relationship between skin compressibility and its stress relaxation behaviour. A

**Table 1.** Details of specimen groups.

| orientation of specimen | strain level (%) | group |
|---|---|---|
| parallel to STL | 5% ($n = 12$) | 1 |
| | 10% ($n = 12$) | 2 |
| | 15% ($n = 12$) | 3 |
| | 20% ($n = 12$) | 4 |
| perpendicular to STL | 5% ($n = 10$) | 1 |
| | 10% ($n = 10$) | 2 |
| | 15% ($n = 10$) | 3 |
| | 20% ($n = 10$) | 4 |

finite-element model based on the AFM observations is developed to understand the nano-level phenomenon responsible for the macro-level compressibility of skin.

## 2. Material and methods

### 2.1. Sample preparation

Skin from the dorsal region of healthy pigs ($n = 2$) (white male porcine Yorkshire breed, age $14 \pm 0.5$ months and weight $120 \pm 5.6$ kg approx.) was collected from the nearest slaughterhouse within an hour of sacrifice. The diet and living environmental conditions for both animals were almost identical. The collection of skin from the single portion (dorsal) minimizes the region-dependent variation in its mechanical properties [18]. Prior to the skin excision, the direction of the spine was marked; later, it was used as a reference to measure the direction of the STL. The hairs and excess fat on the hypodermis were carefully removed using a razor and surgical blade [20,49]. The orientation of STLs from the spine axis was identified using the bulge method. In this supplementary experiment, four circular specimens (30 mm in diameter) with marked spine direction were extracted from the dorsal skin of two different pig animals (two specimens from each animal), later these circular specimens were used in bulge test. The specimens for bulge test were removed from the same skin samples which were used in stress relaxation experiments. The non-uniform dots pattern using a waterproof black India ink was applied on the skin surface (epidermis side). All the specimens were expanded, and the values of apparent linear modulus in 360° directions were calculated using the procedure described in our previous study [25]. The direction corresponding to maximum modulus was marked as STL. Detail of this experiment is provided in electronic supplementary material, §2.1, figure 1S. We observed little variation in the direction of STLs between the animals. Therefore, for presenting the accurate effect of STLs orientation on the calculated mechanical properties of skin, the direction of STLs was not generalized between the animals and marked in each animal individually, and orientation of specimens was decided relative to STLs of individual animal.

After identifying the direction of STLs, rectangular [12,13,50,51] specimens, parallel (48 specimens) and perpendicular (40 specimens) (table 1) to the STL (figure 1a) were cut using the punching machine (SCD Manufacturer, Kolkata, India) and a home-made die (length × width = 75 × 6 mm). The gauge length and width of a typical specimen are shown in figure 1b. All the specimens were subsequently wrapped in phosphate buffer saline (PBS, pH 7.0) and stored at −20°C temperature to preserve their freshness [52]. Prior to the experiment, the specimens were thawed initially at 4°C and then at room temperature. PBS solution was sprayed to avoid dehydration during the thawing period. In this study, the effect of tissue storage was not considered, as all the tests were performed within a week from the date of sacrificing [52]. The thickness of each specimen was measured at four different locations over the gauge length using a laser distance measurement tool (ILD1320-10, Micro-Epsilon, Germany with 0.001 mm resolution). Further, the mean value of thickness was used for the stress calculation.

### 2.2. *Ex vivo* experimental set-up

All *ex vivo* experiments were performed in Servopulser Servo Dynamic System (EHF-L series, Shimadzu, Japan) configured with a 450 N load cell (Honeywell, USA) coupled with two digital cameras (resolution:

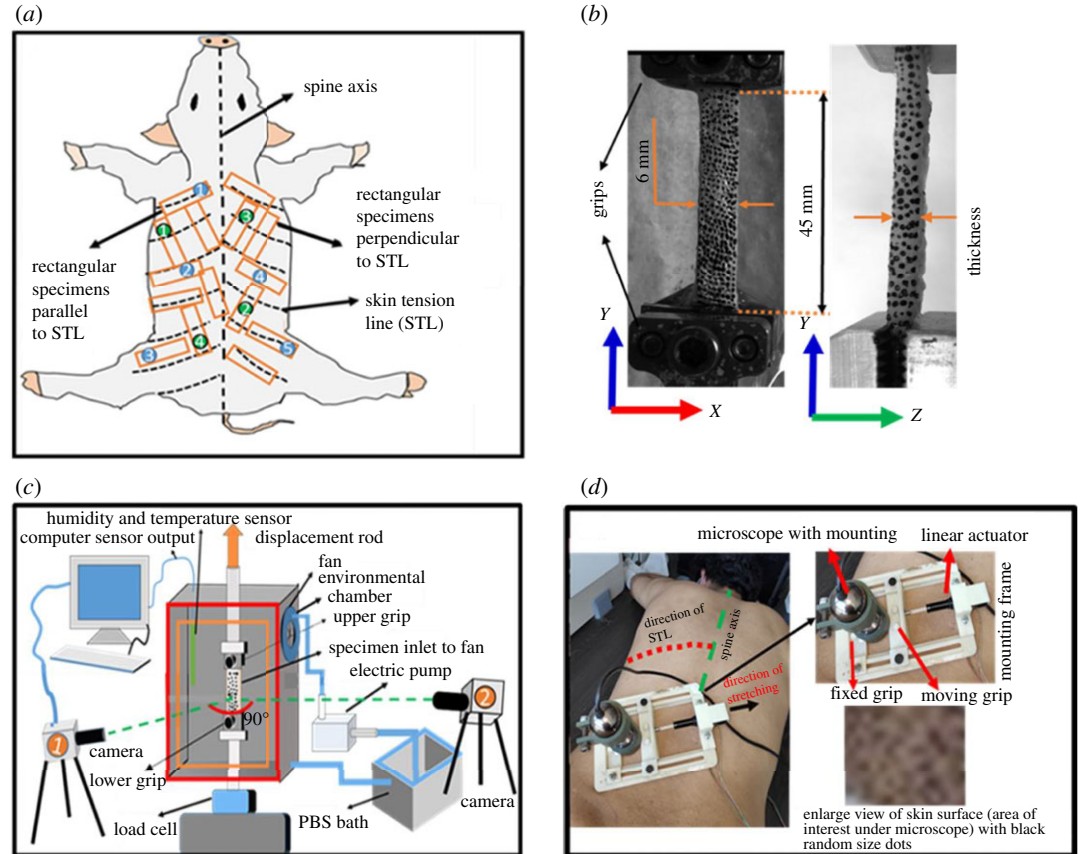

**Figure 1.** (*a*) Schematic shows the rectangle specimens extracted from the dorsal portion of the pig skin with marked orientations to skin tension line (STL). Circle numbers 1–5 in blue circles represent the random locations of five specimens that were excised parallel to STLs, whereas circle numbers 1–4 in green circles represent the random locations of four specimens that were excised perpendicular to STLs. These specimens were used for microstructural assessment. (*b*) A typical specimen with dimensions (gauge length and width) and random-sized black dots. (*c*) A schematic of an experimental set-up coupled with the DIC system, where two identical cameras were mounted perpendicular to each other. Camera 1 captures the deformation of the skin in *X*–*Y* plane, and camera 2 captures deformation in the *Y*–*Z* plane. (*d*) *In vivo* stretching experiment on human back skin. Note: here, the direction of stretching and direction perpendicular to stretching were nomenclatures as *yy* and *xx*, respectively.

5 MP with a frame rate of 15 images s$^{-1}$, Flir Systems Inc., Canada). These cameras were mounted perpendicular to each other and perpendicular to the skin surfaces (figure 1*c*). The specimen was mounted on the machine using customized screw grips with internal daub surfaces [14]. The extra care was taken to avoid the slip of specimens from the grips by gluing the specimen ends (low viscosity instant adhesive, Permabond, Loctite 495) with sandpaper. All experiments were performed inside the home-made environment chamber, where physiological-like conditions were maintained. The relative humidity (RH) was controlled by a small fan with a PBS-soaked sponge. The sponge was kept wet by the continuous supply of PBS using an electric pump. During the test, the environmental conditions inside the chamber were monitored continuously using humidity and temperature sensor DHT 22 (humidity range: 0–100% and accuracy: ±2% RH, temperature range −40°C to 80°C and accuracy ±0.5°C) [25]. A schematic illustration of the test set-up is demonstrated in figure 1*c*.

## 2.3. Digital image correlation and microscopy

A DIC technique was used to measure the strain field in *X*–*Y* (parallel to the skin surface) and *Y*–*Z* (plane of thickness measurement) planes. The surface of specimens (epidermis side) was speckled with a non-uniform dots pattern using a waterproof black India ink (figure 1*b*) [11,25]. These dots were tracked during the stretching and stress relaxation periods of the skin specimen. VIC snap 9.0 software (Correlated Solutions, USA) was used for image capturing, and the full-field strain was measured through the post-processing of captured images in VIC-2D software (Correlated Solutions, USA). The

refraction due to glass (chamber windows between specimen and cameras) may affect the displacement accuracy, but a consistent shift in the speckles does not induce an error in the strain value. A procedure described in our previous study [25] was used to measure the error in the strain value obtained through DIC. This procedure stimulates the rigid body motion and obtains the strain error. The value of error for the above-mentioned experimental set-up was found to be ±0.00181%.

For confirming the repeatability of obtained results, the accuracy of the DIC method was examined by performing a similar experiment on the samples of layered thermoplastic rubber with known and consistent mechanical properties. The obtained results are provided in electronic supplementary material, figure 2S(a–e). These results show a very small error (approx. 1.2%) in the strain value, which confirms the accuracy of the DIC system.

Further, during stretching of the specimens, small fluid bubbles were observed on the skin surface, although no detachment of dots was observed during the stretching. Therefore, to confirm the effect of fluid bubbles on strain calculation, we performed a supplementary test on the metal plate. The black dots similar (e.g. random size and space) to the skin specimens were made on the plate surface and then the plate was fixed vertically on the lower grip of tensile test machine (electronic supplementary material, figure 2S(f)). Twenty to thirty images of plate were captured, and subsequently water was sprayed on the surface of the plate (to mimic the formation of fluid bubble on the skin surface) and again 20–30 images of water-moistened surface were captured (electronic supplementary material, figure 2S(g)). Using DIC, the captured images were analysed and the strain was measured for both the conditions (with and without water). The measured values of strain in dry and wet cases were 0.001 and 0.0019, respectively (electronic supplementary material, figure 2S(h,i)). These values of strain can be considered almost equal to zero, which is expected, as in both cases the plate was not deformed. This small error in strain due to water/fluid flow can be neglected during the finite deformation of specimens. Also, on some points of the specimen, a disturbance in correlation was observed due to the formation of water droplets but it did not affect the average value of strain.

Further, two Dino-Lite microscopes with 5 MPa cameras (ANMO Electronics Corporation, Taiwan) were placed perpendicular to each other for capturing the fluid exudation from $X–Y$ and $Y–Z$ planes during ramp and relaxation. For this experiment, specimens ($n = 5$, parallel to STL and $n = 4$, perpendicular to STL) adjacent to the specimens used in stress relaxation test were excised, and subsequently black paint (solvable in water) was applied on the specimen surface (epidermis side). The microscopes were focused on the small area of specimens and specimens were stretched up to different strain values before holding at these strain values. The fluid exudation during the stretch and hold was captured though the replacement and movement of black paint with fluid flow. Note: these specimens were not included in strain calculation, this experiment only provides the evidence of fluid exudation during ramp and hold, and this experiment throughout the manuscript is nomenclature as microscopic experiment.

## 2.4. *Ex vivo* loading and stress relaxation

Forty-eight specimens parallel to STL and 40 specimens perpendicular to STL had been equally divided into four groups (table 1). Before the actual test, each specimen was preloaded to 0.5 N to remove the slackness of the specimen between the grips [20], and then the load had been tare. The parallel and perpendicular specimens of groups 1, 2, 3 and 4 were loaded up to peak strain 5%, 10%, 15% and 20%, respectively, and then each specimen was held for 30 min at this strain level to observe its stress relaxation behaviour. Here, the input displacement to the machine for achieving the required strain level was calculated based on the gauge length (distance between two grips), which was measured using a digimatic vernier caliper (Mitutoyo-500–196 with least count of 0.01 mm, Japan). All experiments were performed in displacement control mode with a constant displacement rate of 0.1 mm s$^{-1}$. The load data during the ramp and stress relaxation phase were recorded with a 50 Hz sampling frequency, whereas the strain data were measured through DIC with 10 fps. No preconditioning was performed to avoid irreversible damage to the tissue [11,20].

## 2.5. *In vivo* experiment on human skin

Identical to the pig skin location (dorsal region), human skin from the left mid-back region was selected for the *in vivo* experiment. Five healthy males (North Indian) (two, 31 years old and three, 30 years old) were included in this study with their signed consent. The experimental protocol was performed as per the institutional ethical guidelines for the non-invasive human study. Following cleaning of skin

(removal of hair, water, etc.), the direction of STL (approx. 45° from the direction of the spine [16,26]) was marked (figure 1d). After cleaning and STL direction identification, the non-uniform dots were applied on the skin surface using waterproof black India ink (figure 1d); these dots were used to track the deformation of the skin. The deformation on the skin surface was applied through an in-house built tensile test set-up (figure 1d), where the skin was stretched up to 4 mm with 0.5 mm s$^{-1}$ displacement rate and then held at this deformation level for 20 s. The later phase of the experiment was designed to capture the lateral deformation of skin during the hold. For capturing the *in vivo* anisotropy, this experiment was performed in parallel and perpendicular directions of STL. The video of the skin deformation was captured using a microscope with 15 fps (resolution: 5 MP: Dino-Lite, Olympus). The images from the video were extracted using Matlab code, and the strain was calculated using the DIC technique (see §2.3).

## 2.6. Micro-CT and image processing

The native arrangement of collagen fibres was investigated using Micro-CT (Phoenix Nanotom S, GE Sensing and Inspection Technologies, Germany), where five specimens parallel to STL (marked blue in figure 1a) and four specimens perpendicular to STL (marked green in figure 1a) (10 mm in length and $4 \times 2$ mm$^2$ in cross-section area) were collected from the dorsal region. The contrast staining of the skin was performed to increase the X-ray absorbance according to the protocol described in the literature [53]. In brief, all specimens were taken up to 70% ethanol through ascending concentrations (30%, 50% and 70%, 20 min for each). After this step, the specimens were transferred to 30% phosphotungstic acid (PTA) (in the ratio of; 30 ml PTA water solution in 70 ml absolute ethanol) and kept for 72 h. After staining, the specimens were transferred to micro-CT. All the specimens were scanned inside the 70% ethanol-filled glass tube (6 mm in diameter), which was mounted in the machine station. Each specimen was scanned at 1.5 µm isotropic voxel size, and a total of 1200 projection images were acquired on the CCD camera using voltage, 45 kV; beam current, 200 µA; exposure time, 750 ms; and frame average of 10. The reconstruction of obtained images was done using Phoenix Datosx 2.0 software (Phoenix Nanotom S, GE Sensing and Inspection Technologies).

Further, the three-dimensional image dataset was imported in Scan IP 7.0 software (Simpleware, Synopsys Inc., USA). A cube of sub-volume ($2.5 \times 2.5 \times 1.5$ mm$^3$) was cropped, and the collagen fibre bundles were separated from the ground matrix using a global threshold method [54]. The value of the threshold was chosen to maximize the collagen fibre bundles visibility and to minimize noise. Further, to measure the orientation of fibre bundles, the mask of fibres was skeletonized using the auto skeleton module. This procedure measures the distance map of the segmented collagen fibres bundles and then performs a thinning of these bundles. This procedure preserves the native topology of the collagen fibres network, including their length and orientation. Further, the centreline statistics tool was used to measure the mean orientation of collagen fibre bundles in the $X$–$Y$ and the $Y$–$Z$ plane.

## 2.7. Scanning electron microscopy and atomic force microscopy

The native arrangement and shape of collagen fibres and fibrils network in the skin were investigated through SEM and AFM. For these two techniques, a similar protocol was followed as described in our previous study on pig dermis [20]. A brief protocol of SEM and AFM is provided in electronic supplementary material (§§3.1 and 3.2). The whole experimental pipeline is summarized in figure 2.

## 2.8. Modelling of the dermis deformation response at collagen fibril scale

Based on the AFM observation of collagen fibrils arrangement in the dermis, a three-dimensional symmetric model was developed. This representative volumetric element (RVE) of skin was modelled with reduced numbers of collagen fibrils and its dimensions (fibril diameter of 0.13 µm and length of 1 µm). The RVE model was constructed in ABAQUS (Dassault System, France), where the three-dimensional circular collagen fibrils were assembled inside the soft matrix ($0.7 \times 0.7 \times 1.5$ µm$^3$). Similar to AFM results, collagen fibrils inside the matrix were assembled in parallel and staggered way. The collagen fibrils were interconnected through spring elements (representation of cross-linker e.g. GAGs). Apart from this, two left-side collagen fibrils were kept unconnected over the certain length. This was done purposely to observe the role of cross-linkers in the collagen fibrils deformation (as we thought that under the tensile force, the cross-linkers pull the neighbouring collagen fibrils relative to each other). The sliding between the collagen fibrils and matrix was allowed by providing finite

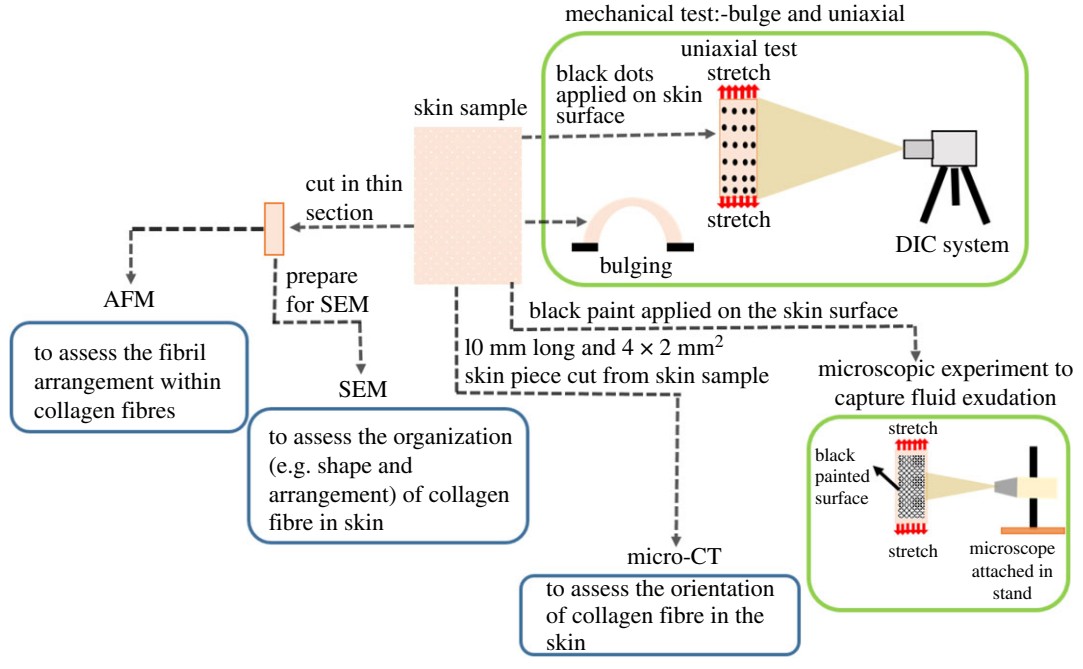

**Figure 2.** The schematic description of experimental pipeline which was used in this study.

sliding-type interaction between matrix and fibrils. Another side, to avoid the penetration of fibrils in the soft matrix during deformation, a hard–normal-contact was defined between the collagen fibrils and matrix surfaces. This model had symmetric surfaces perpendicular to all three Cartesian coordinates to satisfy the Hill's conditions of RVE [55]. For incorporating the infinite size body behaviour during the deformation, the model was simulated under the periodic boundary conditions. A method similar to the work of Pahr & Zysset [56] was used to apply the periodic boundary conditions on the RVE. For finite-element analysis, the values of Young's modulus for collagen fibrils and soft matrix were taken as 1.5 GPa and 0.14 MPa [57,58], respectively. The Poisson's ratio for both these components was assumed 0.3 as its value was found missing in the literature. Further, the stiffness of the interfibrillar cross-linker was taken as $1.3 \times 10^{-3}\,\mathrm{N\,m^{-1}}$ [59]. The model is meshed using the tetrahedron element with an element size of 0.5 µm (electronic supplementary material) and simulated under plane strain conditions. A tensile strain of 0.06 was applied on the RVE along the Z-axis. This applied strain was the part of periodic boundary conditions.

## 2.9. Data analysis

### 2.9.1. Stress relaxation (decay in stress)

The force data during the ramp and hold (stress relaxation period) periods was extracted with time, and corresponding stress values were calculated. Further, the value of the relative change in stress for each specimen during the hold period was calculated using equation (2.1). The mean of this parameter was used for the comparison among the groups.

$$\text{relative change in stress (normalized value)} = \left( \frac{\sigma_0 - \sigma_t}{\sigma_0} \right), \tag{2.1}$$

where $\sigma_0$ and $\sigma_t$ are the value of peak stress and the value of stress during hold at a time ($t$), respectively.

### 2.9.2. Poisson's ratio

DIC was used to capture the longitudinal ($\varepsilon_{yy}$) and lateral ($\varepsilon_{xx}$ or $\varepsilon_{zz}$) strains in the specimens during the ramp and hold periods, and these strains were used to evaluate the Poisson's ratio of the specimens.

$$\text{Poisson's ratio } (\nu_{xy} \text{ and } \nu_{yz}) = -\frac{\ln(1 + \varepsilon_{xx})}{\ln(1 + \varepsilon_{yy})} \text{ and } = -\frac{\ln(1 + \varepsilon_{zz})}{\ln(1 + \varepsilon_{yy})}. \tag{2.2}$$

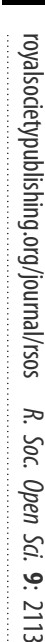

**Figure 3.** (*a*) Ramp and hold portions of experiment at different strain levels. (*b*) Stress–strain curve (average value) of parallel and perpendicular specimens. Map of the engineering strain in *X–Y* (left column) and *Y–Z* (right column) planes at the end of ramp (20% strain level) for (*c*) parallel and (*d*) perpendicular specimens. The black rectangular boxes represent the area of interest over which the average value of strain was measured.

Further, the value of the increase in Poisson's ratio during the relaxation portion was calculated using equation (2.3). Equation (2.2) used the average value of all localized strains across the surface (figure 3) to calculate the average value of Poisson's ratio. The average of localized strains across the surface was calculated over the selected area of interest (AOI) (figure 3) and this AOI was taken far from the boundaries of specimens, which eliminates the boundary effect in strain calculation. The mean value of the increase in Poisson's ratio was used for all the comparisons.

$$\text{Increase in Poisson's ratio} = \left( \frac{(\nu_t)_{xy} - (\nu_0)_{xy}}{(\nu_0)_{xy}} \right) \quad \text{and} \quad \left( \frac{(\nu_t)_{yz} - (\nu_0)_{yz}}{(\nu_0)_{yz}} \right), \tag{2.3}$$

**Table 2.** Mean value of elastic modulus, transitional strain and Poisson's ratio.

| parameters/orientation | | parallel | | perpendicular | | p-value |
|---|---|---|---|---|---|---|
| elastic modulus (MPs) | | 114.00 ± 18.58 | | 65.93 + 10.28 | | <0.01 |
| transition strain (%) | | 4.30 ± 0.82% | | 8.8 ± 0.56% | | <0.01 |
| | | **value of Poisson's ratio** | | | | |
| peak strain | loading point | parallel to STL | | perpendicular to STL | | |
| | | $v_{xy}$ | $v_{yz}$ | $v_{xy}$ | $v_{yz}$ | |
| 20% | initial | 0.423 ± 0.053 | 0.614 ± 0.102 | 0.441 ± 0.038 | 0.521 ± 0.156 | |
| | final | 1.130 ± 0.210 | 1.381 ± 0.351 | −1.710 ± 0.290 | 1.411 ± 0.289 | |

where $(v_0)_{ii}$ ($i = x$, $y$ and $z$) is the average value of Poisson's ratio corresponding to peak stress and $(v_t)_{ii}$ is the average value of Poisson's ratio at time $t$ during the hold period.

## 2.10. Statistical analysis

For the comparison of the obtained experimental results, the statistical analyses were performed using SPSS (21.0, IBM, USA) software. The analysis of variance (ANOVA) test followed by the Tukey–Kramer with *post hoc* test was used to determine the influence of strain level on the value of the relative change in stress and increase in Poisson's ratio during stress relaxation [14,60]. Before performing these tests, the normality of data distribution was evaluated using the Kolmogorov–Smirnov test [60]. The same statistical procedure was also performed to compare the values of Poisson's ratio at different strain points during the loading period. Moreover, a paired Student's *t*-test was implemented to investigate the impact of the orientation of specimens (parallel and perpendicular to STL) on the relative change in stress and increase in Poisson's ratio during stress relaxation. The level of significance for elimination of the false direction was set $p < 0.05$. All the results were reported by mean ± s.d. with the *p*-value (\*\*$p < 0.01$, \*$p < 0.05$). Pearson correlation test was also performed to confirm the significant correlation between the increase in Poisson's ratio and the relative change in stress during stress relaxation. For this test, level of significance was set with *p*-value $< 0.05$.

# 3. Results

## 3.1. Loading portion

Figure 3*a* demonstrates the input profile of ramp and hold phases for different strain levels, and figure 3*b* shows the stress–strain curves (average value) for parallel and perpendicular specimens for different strain levels. For the low value of strain (approx. 1.0–1.5%), the pig skin shows (figure 3*b*) isotropic behaviour and the value of stress in the parallel and perpendicular specimen was not found significantly different. However, beyond this strain value, the orientation-dependent stress–strain response was observed, and the value of elastic modulus (table 2) was found significantly higher in parallel specimens. These results were found in good agreement with the literature [16,25]. Further, figure 3*b* illustrates a larger heal region for perpendicular specimens than parallel specimens, and the transition strain was also found significantly larger for perpendicular specimens (electronic supplementary material, figure 3S(a,b); and table 2). This orientation-dependent behaviour indicates the anisotropic nature of the skin, which is a result of different underlying mechanisms of collagen fibres deformation [12] and their initial orientation [16].

Figure 3*c,d* shows the strain maps (obtained through DIC corresponding to 20% strain level) for both parallel and perpendicular specimens, respectively, in the X–Y and Y–Z planes. The strain maps corresponding to strain levels of 5%, 10% and 15% are given in electronic supplementary material,

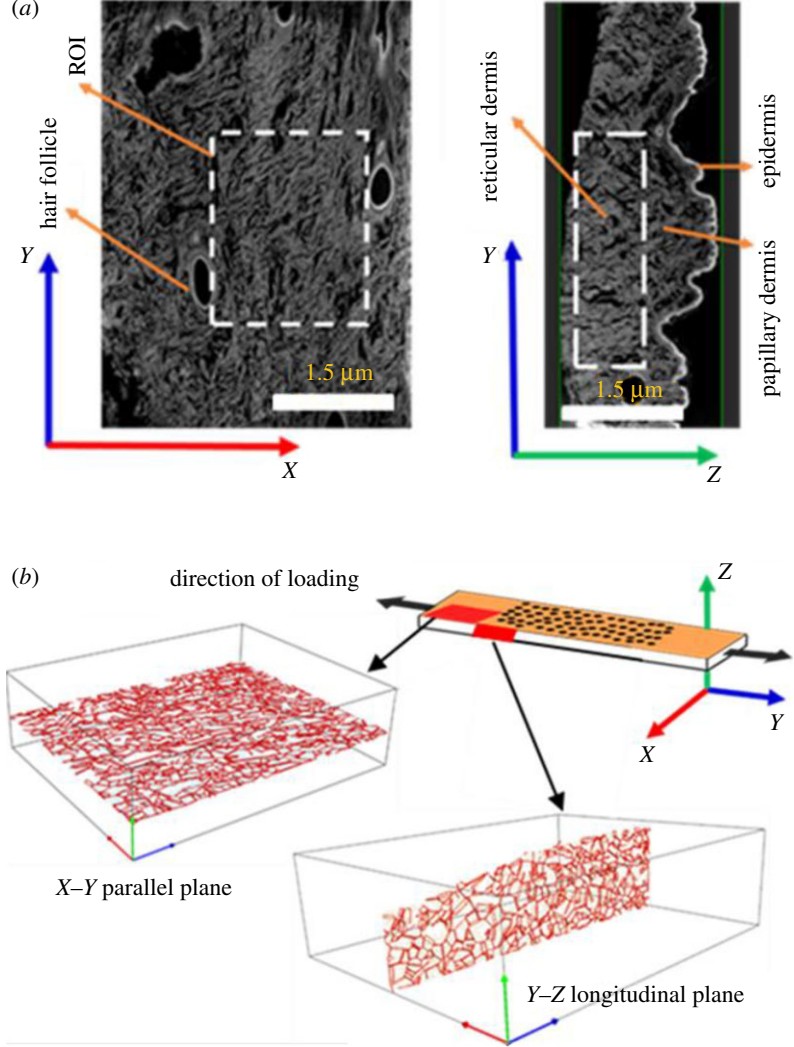

**Figure 4.** (*a*) Micro-CT slices of the skin in *X*–*Y* (left) and *Y*–*Z* (right) planes. The white rectangle represents an area of interest. (*b*) The upper figure represents the schematic of sections obtained from the dermis in *X*–*Y* and *Y*–*Z* planes, and the bottom figures present the post-processed images of skin slices.

figure 3S(c–f). The same figures show the large variation in strain distribution across the boundaries of specimen, which could be due to the boundary effect. For eliminating the boundary effect during the average strain calculation, the AOI was selected away from the specimen boundaries. The value and distribution of $\varepsilon_{zz}$ on the *Y*–*Z* plane were found almost similar in both orientations, which is quite obvious as thickness direction will remain the same irrespective of STL. Also, the average values of $\varepsilon_{yy}$ in both planes (*X*–*Y* and *Y*–*Z*) were found almost the same; however, the distribution of $\varepsilon_{yy}$ in the *Y*–*Z* plane appeared more heterogeneous. The heterogeneity of strain in this plane could be the result of the non-uniform thickness (figure 4*a*,*b*) over the gauge length. Further, under stretching, parallel specimens show lateral contraction ($\varepsilon_{xx} < 0$ and $\varepsilon_{zz} < 0$) in both planes, whereas perpendicular specimens show lateral expansion ($\varepsilon_{xx} > 0$) in *X*–*Y* plane and contraction ($\varepsilon_{zz} < 0$) in *Y*–*Z* plane.

The values of the Poisson's ratio ($\nu_{xy}$ and $\nu_{yz}$) were calculated using equation (2.2) and are given in table 2. The variation of Poisson's ratios ($\nu_{xy}$ and $\nu_{yz}$) in parallel and perpendicular specimens with the longitudinal strain ($\varepsilon_{yy}$) is demonstrated in figure 5*a*–*c*. For parallel specimens, a positive linear correlation was found between $\nu_{xy}$ and $\varepsilon_{yy}$ ($R^2 = 0.9821$), and $\nu_{yz}$ and $\varepsilon_{yy}$ ($R^2 = 0.9735$). A similar trend of $\nu_{yz}$ was also observed in the perpendicular specimens (electronic supplementary material, figure 3S(g)), whereas value of $\nu_{xy}$ in perpendicular specimens was found decreasing nonlinearly (initially fast and then slow) with $\varepsilon_{yy}$. The values of Poisson's ratios for parallel and perpendicular specimens are provided in table 2, which are outside the limit of incompressible materials. These values were found in good agreement with the reported value of Poisson's ratio for skin, tendon and ligament [11,43,45].

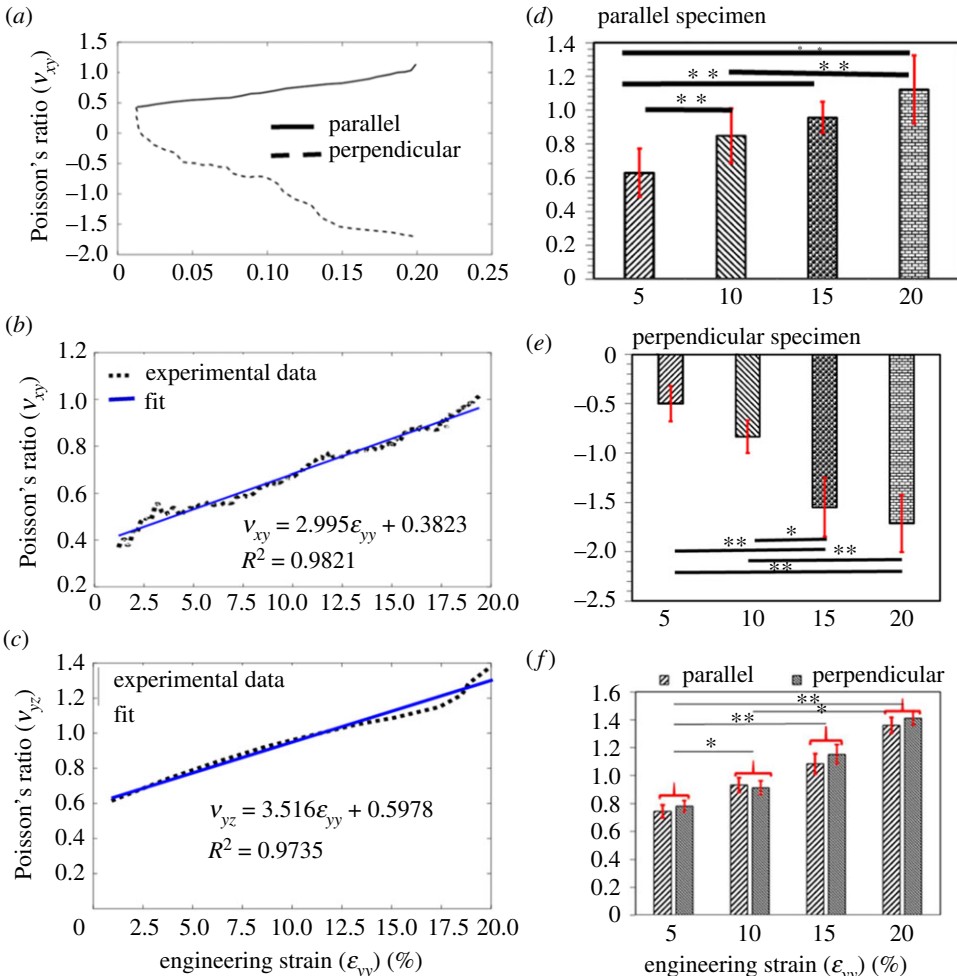

**Figure 5.** (a) The trend of Poisson's ratio (average value) for parallel and perpendicular specimens when plotted against applied strain. The linear correlation between Poisson's ratio (b) $\nu_{xy}$ and (c) $\nu_{yz}$ and applied strain for parallel specimens. Note: in perpendicular specimens, $\nu_{xy}$ and applied strain were not found correlated; however, the trend for $\nu_{yz}$ was found similar to parallel specimens (electronic supplementary material, figure 3S(g)). Comparison of Poisson's ratio ($\nu_{xy}$) (X–Y plane) among the different strain levels in (d) parallel specimens and (e) perpendicular specimens. (f) Comparison of Poisson's ratio ($\nu_{yz}$) (Y–Z plane) among different strain levels in parallel and perpendicular specimens. Note: $^*p < 0.05$ and $^{**}p < 0.01$.

Figure 5d–f shows the comparison of Poisson's ratio among the applied strains for parallel and perpendicular specimens. For the parallel specimens, a significant difference in the values of the Poisson's ratio ($\nu_{xy}$ (figure 5d) and $\nu_{yz}$ (figure 5f)) was found between strain levels of 5% and 10%; 5% and 15%; 5% and 20%; 10% and 20%; however, these values were not found significantly different between strain levels of 15% and 20%. A similar trend was also observed for $\nu_{yz}$ (figure 5f) in perpendicular specimens, whereas the values of $\nu_{xy}$ (figure 5e in perpendicular specimens were found significantly different between strain levels of 5% and 15%; 5% and 20%; 10% and 15%; and 10% and 20%.

## 3.2. Stress relaxation portion

This section presents the results of relative change in stress (calculated from equation (2.1)) and increase in Poisson's ratio (calculated from equation (2.3)) during the stress relaxation phase. The electronic supplementary material, figure 4S(a–i) shows the trend of relative change in stress and increase of Poisson's ratio during the stress relaxation for parallel and perpendicular specimens. The subfigures of electronic supplementary material, figure 4S(a–i) and (a–j) illustrate the continuous decrease in the values of $\varepsilon_{xx}$ and $\varepsilon_{zz}$ during the stress relaxation phase.

In electronic supplementary material, figure 4S(a–h), three distinct phases of stress relaxation can be observed; (i) primary phase (approx. 0–150 s) with a very fast rate of relative change in stress; (ii)

secondary phase (approx. 150–400 s) with the moderate rate of reduction in stress; (iii) tertiary phase (approx. 400–1800 s) with a very slow rate of reduction in stress. This trend of relative change in stress was found similar in both orientations for all applied strains except for the 5% strain in parallel specimens, where the relative change in stress became almost zero after approximately 400 s of hold. Furthermore, in parallel specimens, a fast increase in the value of Poisson's ratios ($v_{xy}$ and $v_{yz}$) was observed during the initial approximately 400 s of the stress relaxation, and afterwards, the increase in the values of $v_{xy}$ and $v_{yz}$ became almost zero. This trend was also found similar in the perpendicular specimens for the value of $v_{yz}$; on the other hand, the value of $v_{xy}$ in perpendicular specimens was found increasing with constant rate during the whole relaxation period (electronic supplementary material, figure 4S(b,d,f and h)). Further, for parallel specimens, the relative change in stress and increase in Poisson's ratio ($v_{xy}$) were found significantly larger ($p < 0.01$, $p < 0.05$) for 5% hold strain than 10%, 15% and 20% hold strains (figure 6a,c). The relative change in stress was also found significantly larger ($p < 0.05$) for 10% hold strain than 20% hold strain; however, the value of the increase in Poisson's ratio was not found significantly different between these two hold strains. Conversely, for perpendicular specimens, the relative change in stress and increase in Poisson's ratio were found significantly larger for 15% and 20% hold strains than for 5% and 10% hold strains ($p < 0.05$, $p < 0.01$). These two parameters were also found larger for 20% hold strain than 15% hold strain (figure 6b,d). Also, during the stress relaxation, the increase in Poisson's ratio in the $Y$–$Z$ plane for both orientations was found significantly large for low hold strain (5%) (figure 6e). Note: the distribution of longitudinal and lateral strains before and after the stress relaxation is presented in electronic supplementary material, figure 5S. Further, the results of the Pearson correlation test showed a strong correlation between relative change in stress (stress relaxation) and an increase in Poisson's ratio of parallel specimens. The $R$-value from this analysis were found 0.94 ($p < 0.0001$), 0.8533 ($p < 0.0001$), 0.8331 ($p < 0.0001$) and 0.7931 ($p < 0.001$) corresponding to 5%, 10%, 15% and 20% hold strain, respectively. For the perpendicular specimens, a significant correlation was not found between the relative change in stress and the increase in Poisson's ratio; however, Poisson's ratio was observed increasing during the stress relaxation phase.

Moreover, the obtained results from stress relaxation experiments illustrate that the viscoelastic behaviour of skin was direction dependent. The relative change in stress after 30 min at 5% and 10% strain level was found significantly larger ($p < 0.01$, $p < 0.01$) for parallel specimens than perpendicular specimens. Whereas, for hold strains of 15% and 20%, this parameter was found significantly larger ($p < 0.05$, $p < 0.01$) for perpendicular specimens than the parallel specimens; these results were found in good agreement with the literature [11,38]. Also, this direction-dependent relative change in stress value was found in correlation with the increase in the Poisson's ratio. Further, the increase in Poisson's ratio ($v_{xy}$) corresponding to 5% and 10% hold strains were found significantly larger ($p < 0.01$) for parallel specimens than perpendicular specimens and vice versa for 15% ($p < 0.05$) and 20% ($p < 0.01$) hold strains. The increase in Poisson's ratio ($v_{yz}$) was found almost the same in both orientations for all hold strains (figure 6e), which was quite evident as the direction of thickness will remain the same irrespective of STL. This correlation between Poisson's ratio and stress relaxation was reported herein, the first time, to our knowledge. This orientation-dependent mechanical response could be the result of the dispersed orientation of collagen fibres in the dermis.

## 3.3. Result of *in vivo* experiment

Figure 7a–d demonstrates the longitudinal ($e_{yy}$) and lateral ($e_{xx}$) strain maps on the human skin. These results are presented for $X$–$Y$ plane (plane parallel to the epidermis). About $39.20 \pm 1.41\%$ (strain) lateral contraction of skin was observed when it was stretched up 40% strain along the STLs (figure 7e), however, for same value of applied strain, the lateral contraction of skin was about $10.80 \pm 1.23\%$ when it was stretched perpendicular to STLs (figure 7f). Similar to the *ex vivo* study, the value of Poisson's ratio was also found to increase with the increase of applied strain when skin was stretched parallel to STL. The value of Poisson ratio corresponding to 40% of applied strain (stretched along the STLs) was found about $1.46 \pm 0.13$ (subfigure of figure 7e). On the other hand, when the skin was stretched perpendicular to STL, the value of Poisson's ratio was found $0.36 \pm 0.05$ (subfigure of figure 7f). Furthermore, during the hold, the value of longitudinal strain was found almost constant, whereas the value of lateral strain was found to decrease with time (figure 7e,f). The values of Poisson's ratio at the end of the hold for parallel and perpendicular stretching to STLs were $1.92 \pm 0.23$ (figure 7e) and $0.49 \pm 0.062$ (figure 7f), respectively.

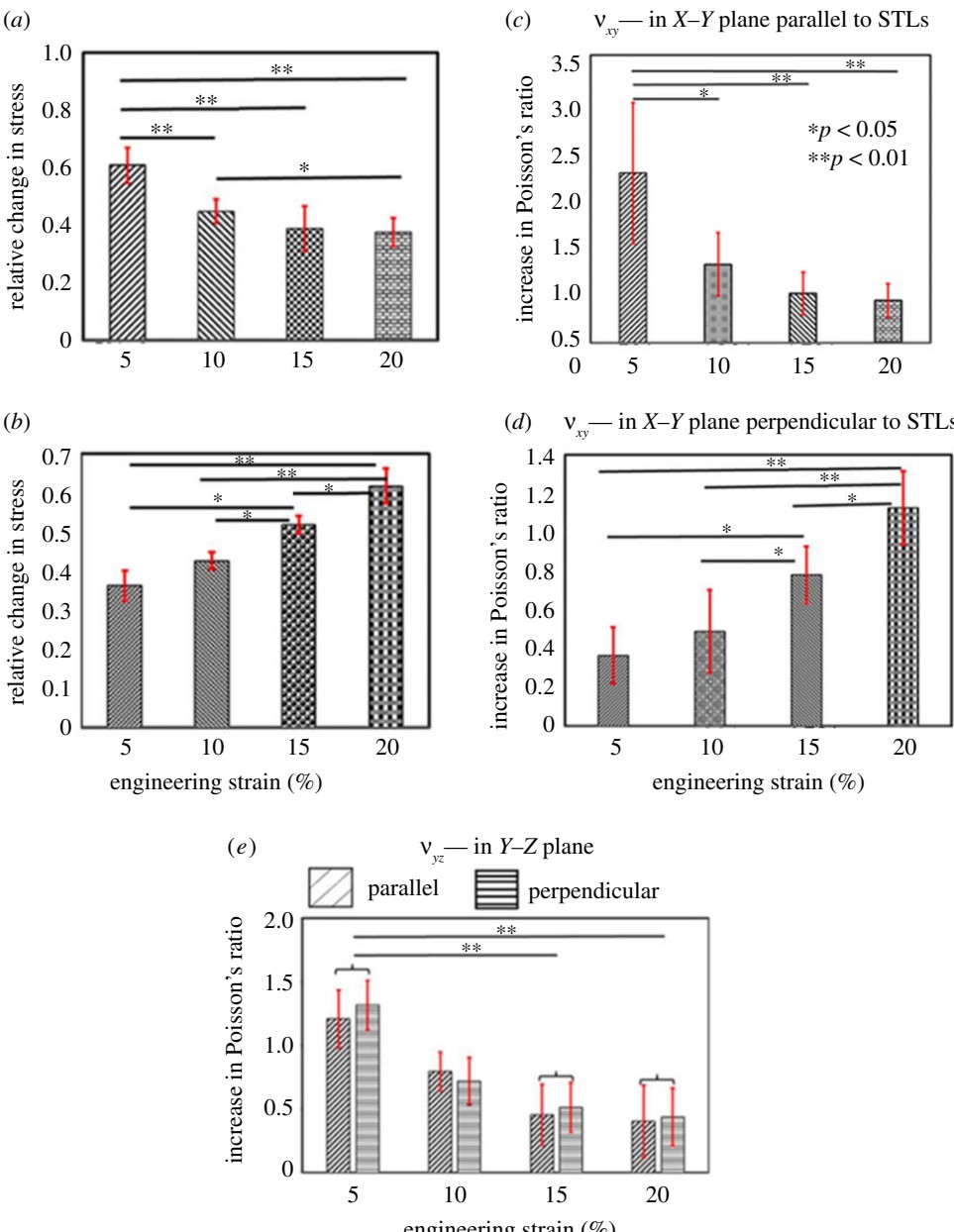

**Figure 6.** Comparison of relative change in stress (mean ± 1 s.d.) among different hold strains (e.g. 5%, 10%, 15% and 20%) for (*a*) parallel specimens and (*b*) perpendicular specimens. Comparison of increase in Poisson's ratio in *X–Y* plane ($\nu_{xy}$) during stress relaxation for (*c*) parallel and (*d*) perpendicular specimens. (*e*) Comparison of increase in Poisson's ratio in *Y–Z* plane ($\nu_{yz}$) during stress relaxation for parallel and perpendicular specimens. Note: $^{*}p < 0.05$ and $^{**}p < 0.01$.

## 3.4. Collagen fibre orientation, collagen fibril and fibre arrangement

The orientation of collagen fibres from the loading direction was calculated as an angle between the collagen fibres principal axis and direction of loading (figure 3*b*). The histograms in figure 8*a–d* display the density distribution of collagen fibres for a 180° angular span. In parallel specimens, the mean orientation of collagen fibres from the loading direction (figure 8*a,c*) was found 2.4° in the *X–Y* plane and 7.5° in the *Y–Z* plane. Whereas for the perpendicular specimens, the mean orientation of collagen fibres from the loading direction in the *X–Y* and *Y–Z* planes was found 83° and 12.5°, respectively (figure 8*b,d*). These results were found in good agreement with the previously published literature [11,16,25,48,61].

Further, figure 9*a,b* demonstrates the AFM images of collagen fibrils in the *X–Y* and *Y–Z* plane. These results showed the parallel arrangement of collagen fibrils within the collagen fibre. The obtained value

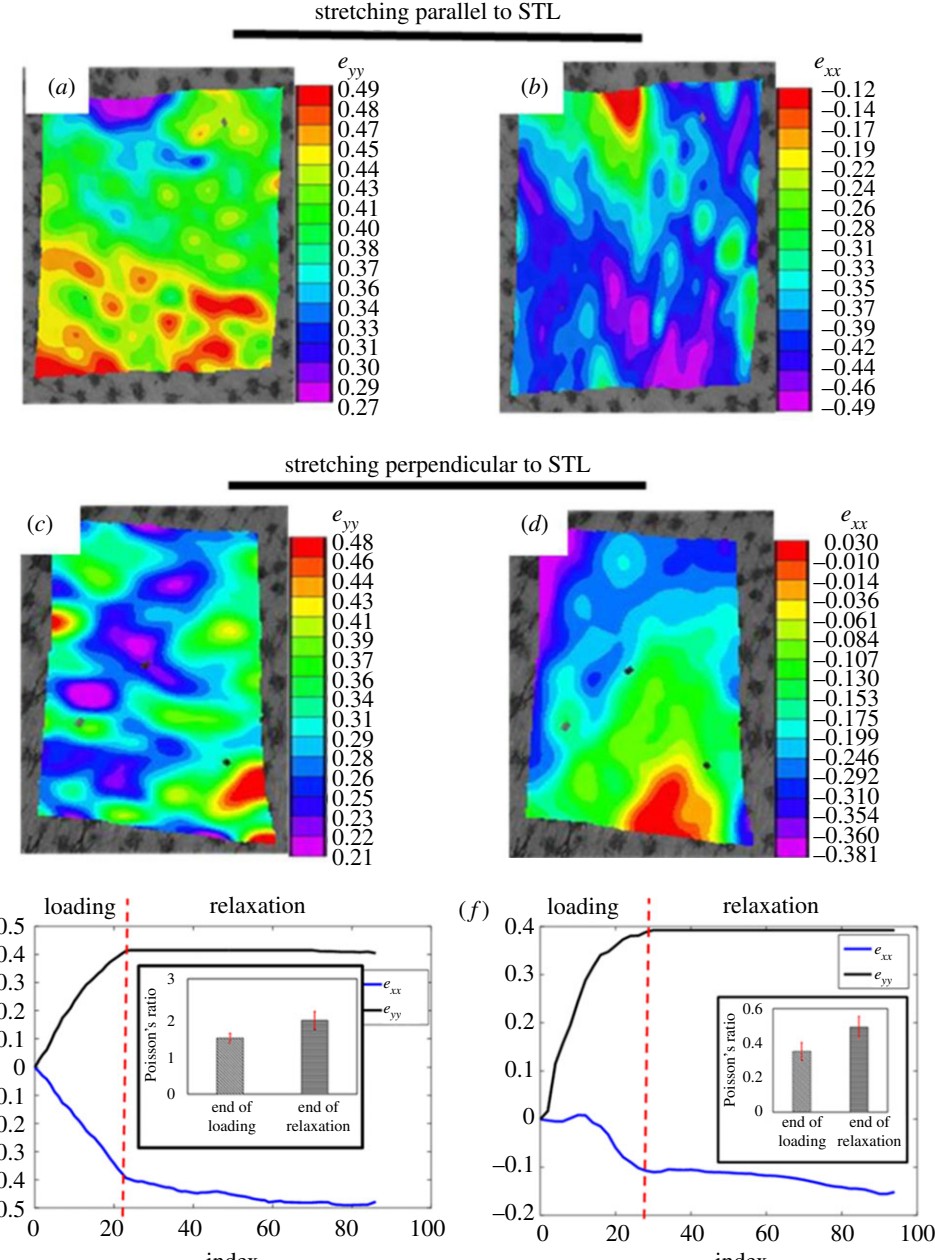

**Figure 7.** Map of engineering strain when human skin is stretched (*a*,*b*) parallel and (*c*,*d*) perpendicular to STL The trend of engineering strains ($\varepsilon_{xx}$ and $\varepsilon_{yy}$) when stretched (*e*) parallel and (*f*) perpendicular to STL.

of D-periodicity and fibril diameter were found $66.0 \pm 2.50$ and $130.0 \pm 10.52$ nm, respectively. This obtained value collagen fibrils diameter was used in finite element analysis (FEA) simulation. The SEM micrograph showed thick bundle of undulated collagen fibres in the pig dermis that were found randomly organized (figure 10*a*). From the observations of SEM we proposed an analogy of collagen fibre arrangement using white wrinkled papers (figure 10*b*,*c*), which was later used to understand the effect of collagen fibres arrangement on the deformation response of skin. The uniaxial stretching of wrinkled white paper showed the unfolding and rotation of wrinkles which leads to lateral expansion of paper (figure 10*d* and electronic supplementary material, video-2).

## 3.5. Exudation of fluid from skin specimen

The results obtained through microscopic experiments are illustrated in electronic supplementary material, videos-1 and -3(a and b). These results showed the formation of water layer on the skin surface when it was uniaxially stretched. The formation of water bubbles was observed to increase (as

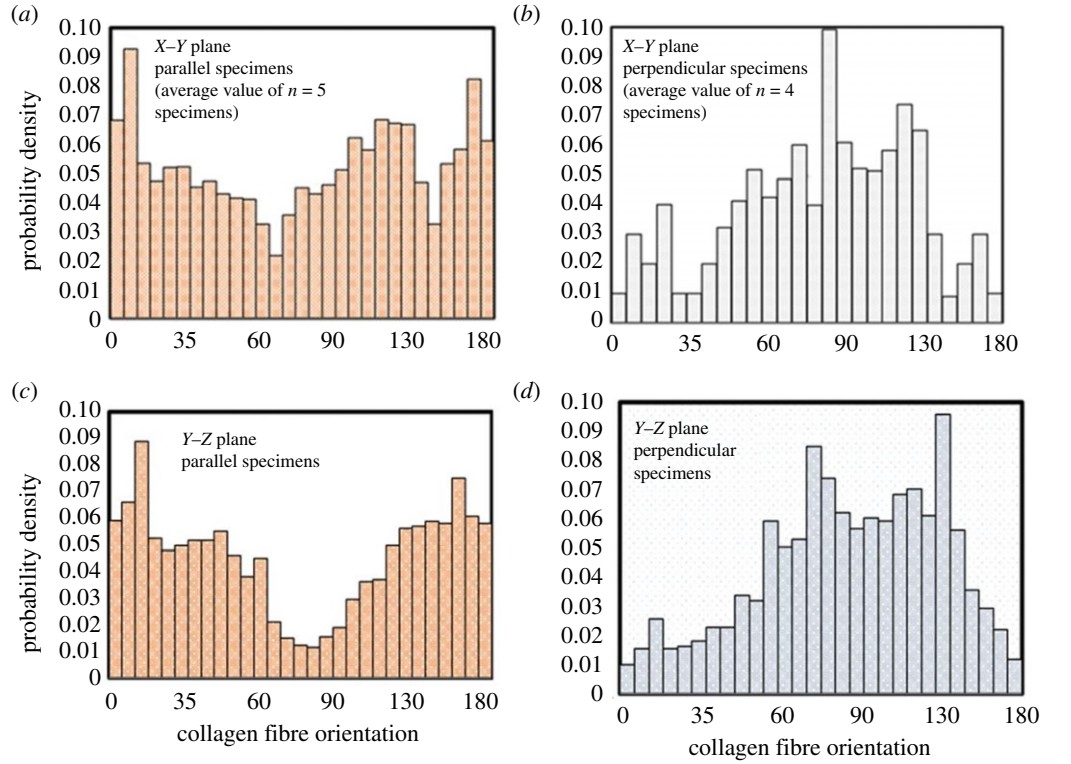

**Figure 8.** Histograms of collagen fibres orientations in X–Y plane (a) parallel specimens and (b) perpendicular specimens, and in Y–Z plane (c) parallel specimens and (d) perpendicular specimens. Note: collagen orientation is presented in half hemisphere of 180°.

black paint was displaced/replaced more corresponding to the large value of strain) with applied stretch. Also, microscopic experiments showed flow of fluid on the skin surface during the hold period of experiments (electronic supplementary material, video-3(a and b)). These results confirmed the exudation of fluid from the tissue during the ramp and stress relaxation period of specimens.

## 3.6. Finite element analysis simulation

Figure 11a demonstrates the volumetric RVEs of skin, where collagen fibrils were assembled within the soft matrix. The same model with tetrahedron meshing is provided in electronic supplementary material, figure 7S(a). Figure 11b presents the cross-sectional view of RVE at undeformed state; three-dimensional view corresponding to this state is provided in electronic supplementary material, figure 7S(b). Another side, figure 11c,d and e shows the deformed cross-sectional view of RVE corresponding to applied strain of 0.02, 0.04 and 0.06, respectively. The deformed three-dimensional views of RVE corresponding to different value of applied strains are given in electronic supplementary material, figure 7S(c,d). The FEA results showed that the strain applied to the ends of RVE transferred between the fibrils through stretching of spring elements (cross-linkers) and shearing between fibrils and matrix, the latter occurred due to the relative sliding of fibrils and matrix. Further, the applied stretch on RVE induced a tensile force on spring elements which pulls neighbouring interconnected fibrils relative to each other. Another side, unconnected collagen fibrils did not come close due to lack of pulling force generated by cross-linker. Further, FEA results showed that centre-to-centre distance of neighbouring collagen fibrils was found to decrease with increase in applied strain. This relative pulling of neighbouring fibrils induced a compressive strain on the encapsulated soft matrix, and its value was found to increase with increase in applied strain (electronic supplementary material, figure 7S(e)). The induced compressive strain squeezed the encapsulated soft matrix (figure 10c–e).

## 4. Discussion

This work explores the correlation of skin microstructure with its three-dimensional deformation response and stress relaxation behaviour. The understanding about the stress relaxation behaviour of

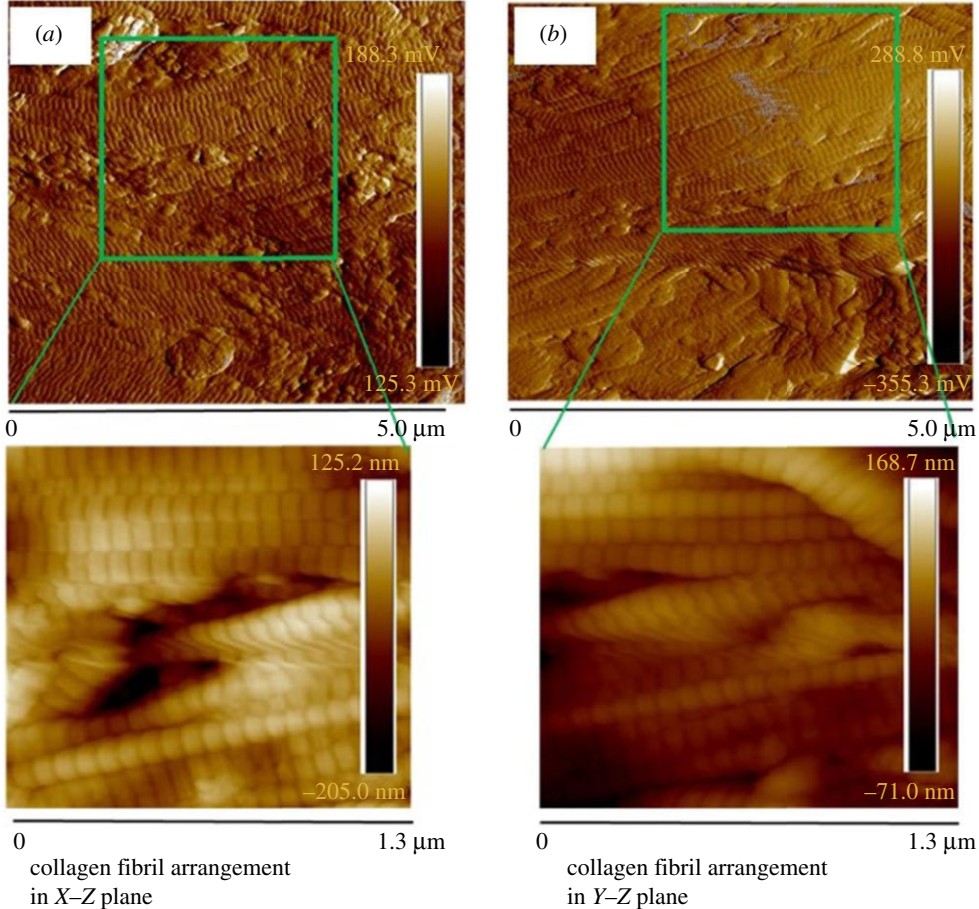

**Figure 9.** AFM images of collagen fibrils configuration in (*a*) *X–Y* and (*b*) *Y–Z* planes. High magnification images show the almost parallel arrangement of collagen fibrils within the collagen fibres. This trend was found similar in both the planes.

the skin will help in large wound closure [14], where a short period of stress relaxation before the completed closure of a large wound will reduce the amount of stress on the skin in the vicinity of the suture as well as on the suture itself. This can reduce the risk of necrosis, pressure ischaemia, skin tear and suture failure due to skin stretching [31].

For parallel specimens, the large values of Poisson's ratios $v_{xy}$ and $v_{yz}$ (>0.5, beyond the limit of incompressibility) indicates the loss in the volume of skin. Quantitatively, about 25% volume of the tissue is lost upon stretching to 20% strain. This loss in volume of the tissue is the result of fluid effluxion from the tissue, which is confirmed through the formation of the fluid droplet on the surface of the skin (electronic supplementary material, video-1, for the detail of method, electronic supplementary material, §4.0).

Further, during stretching, the cause of fluid effluxion from the skin can be explained through the deformation behaviour of collagen fibrils within the soft matrix. For this purpose, a three-dimensional volumetric RVE was created and FEA was performed in this RVE. The RVE model was created based on AFM images of collagen fibrils where a parallel arrangement of collagen fibrils is observed with the collagen fibres (figure 9*a,b* and electronic supplementary material, figure 6S). This trend of collagen fibrils arrangement is found true in both planes (*X–Y* and *Y–Z*). Furthermore, our previous study on the pig dermis [20] and the work of Yang *et al.* [12] on the white rabbit skin revealed that the strain accumulation in collagen fibrils was less than the applied strain. From these observations, it is concluded that the load transfer between the collagen fibrils is due to interfibrillar sliding. All these pieces of information have been incorporated in the RVE model (figure 11*a*). The FE results of this model show that under stretching, the collagen fibrils shift toward each other due to pulling force exerted by the interfibrillar cross-linker (GAGs) (figure 11*b–e*). The relative pulling of neighbouring fibrils toward each other induced a compressive strain, which results in the squeezing of interfibrillar (encapsulated) matrix (figure 11*b–e*). This deformation phenomenon can generate a positive pressure inside the interfibrillar matrix, which may lead to pushing of IF outward from the tissue. Further, the

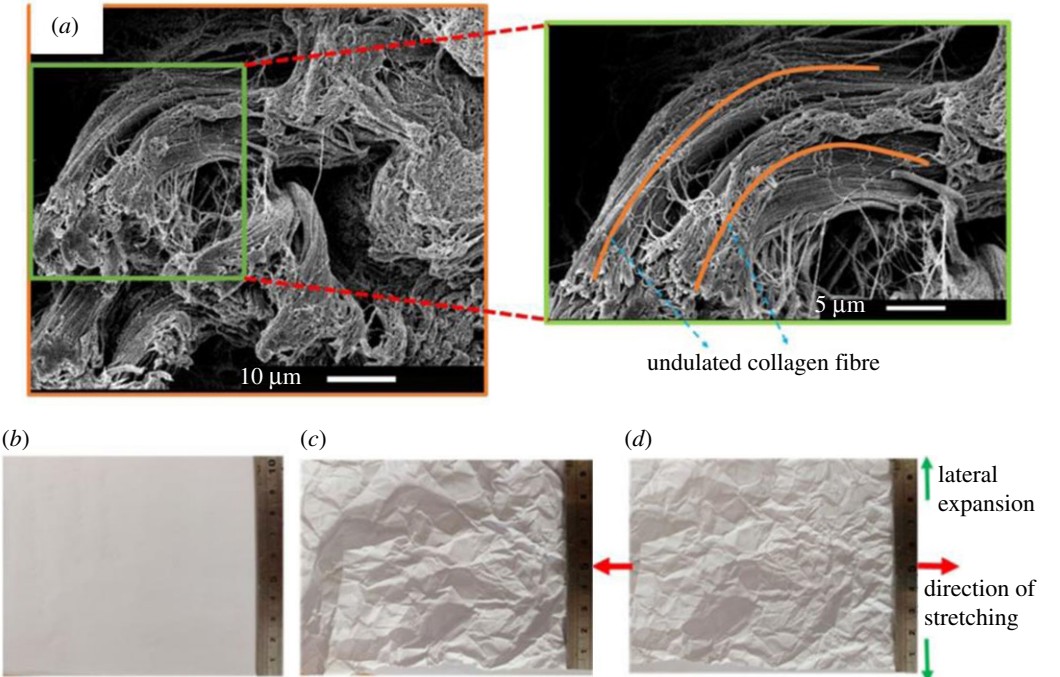

**Figure 10.** (*a*) SEM micrograph of collagen fibre. High magnified micrograph of collagen fibres in pig dermis, superimposed by the simplified representation of curved collagen fibres. Photograph of white paper (*b*) without wrinkles, (*c*) with undeformed wrinkles (*d*) with deformed wrinkles (movie of white paper deformation is provided in electronic supplementary material, video-2).

increase in compressive strain with an increase in the applied strain (electronic supplementary material, figure 7S(e)) can be a cause of large skin volume loss at a high strain. This finite element method (FEM) simulation provides the reasoning for the increase in Poisson's ratio with applied strain but does not predict the volume loss of the tissue. However, for validating the above-stated hypothesis, a more accurate finite-element analysis with incorporation of poroelastic properties of tissue is required in future, as this model does not capture the loss in fluid from the tissue.

Another side, this hypothesis does not seem to be true for the perpendicular specimens where the collagen fibres were oriented at 83° (*X–Y* plane) away from the loading direction. Therefore, stretching of specimens in the direction which is far from the direction of collagen fibres orientation results in large rotation and lateral delamination of collagen fibres. The large rotation of collagen fibres in perpendicular specimens can be confirmed through the higher value of transition strain in perpendicular specimens (electronic supplementary material, figure 3S(a,b); and table 2), whereas their delamination can be observed through SEM (electronic supplementary material, figure 8S(b)). This deformation response of the collagen fibres limits the stretching of collagen fibrils which leads to negligible or very less IF exudation from the tissue, particularly for the low value of applied strain.

Furthermore, the different lateral deformation behaviour of the skin (in the *X–Y* plane) between parallel (lateral contraction) and perpendicular (lateral expansion; negative Poisson ratio) specimens indicates its direction-dependent auxetic behaviour. This anomalous behaviour of skin is found to be related with the native configuration of collagen fibres; in parallel specimens, the collagen fibres are oriented 2.4° from the loading direction, whereas in the perpendicular specimens, the orientation of collagen fibres is 83° degrees from the loading axis (figure 8*a,c*). In addition to this, SEM micrographs (figure 10*a*) show the undulated shape of collagen fibre in the undeformed skin. During the stretching of skin, the undulated collagen fibres undergo rotation, bending and straightening, simultaneously [12], and it can be considered that the amount of rotation of collagen fibres depends on the value of their orientation from the loading direction. Therefore, the large rotation of collagen fibres along with their bending and straightening can be the cause of lateral expansion (auxeticity) in perpendicular specimens. This speculation can be explained with the help of wrinkled white paper (figure 10*b–d*), where the stretching of wrinkled paper results in its lateral expansion due to the rotation and unfolding of wrinkles (the deformation of wrinkled paper is demonstrated in electronic supplementary material, video-2). Furthermore, from a mechanics point of view, the cause of lateral expansion can be explained with the help of the combined deformation response of collagen fibres

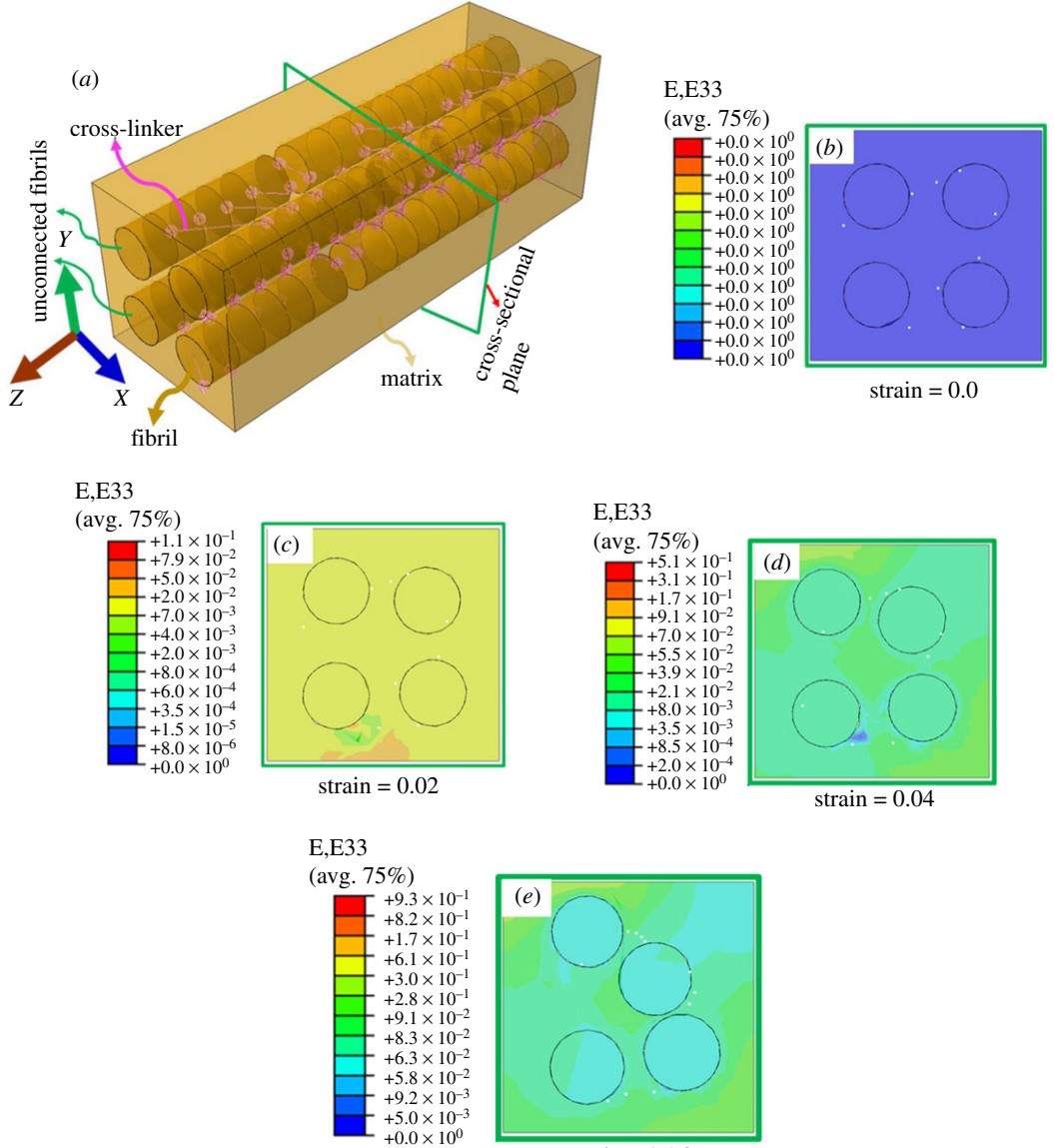

**Figure 11.** (*a*) The representative volumetric element (RVE) model of collagen fibril and matrix assembly. The collagen fibrils assembled in parallel and staggered manner inside the soft matrix. The RVE with tetrahedron mesh is presented in electronic supplementary material, figure 7S(a). (*b*) A cross-sectional view RVE at undeformed state. Cross-sectional view of RVE at deformed states corresponding to applied strain (*c*) 0.02 (*d*) 0.04 (*e*) 0.06. The three-dimensional view of RVE in deformed state is provided in electronic supplementary material, figure 7S(c,d). All neighbouring collagen fibrils except unconnected fibrils (two left side) come close to each other under the stretching of RVE.

and ground matrix. For this hypothesis, two symmetric families of collagen fibres are considered [61] that are embedded in an isotropic ground matrix and confined in the plane. In the first configuration, the mean orientation of collagen fibres is taken close to the loading direction (similar to the collagen fibre orientation (2°) in parallel specimens). Whereas in the second configuration, the mean orientation of collagen fibres is taken far from the loading direction (similar to the collagen fibre orientation (83°) in perpendicular specimens). When the stretch is applied in the collagen fibre-matrix assembly of the first configuration, the component of fibre tension in the perpendicular direction has a lower value than the component of fibre tension parallel to the loading direction. These results are due to the small value of angle (say $\alpha = {\sim}2°$) (figure 8*a*,*c*) between the mean axis of collagen fibres and the loading direction. Therefore, the small value of lateral tension cannot be able to resist the lateral contraction of the ground matrix, and the net result leads to lateral contraction of the skin. Conversely, when the stretched is applied in the second configuration of the collagen fibre-matrix assembly, the component of fibre tension perpendicular to the loading direction will be more than the component of

**Table 3.** Mean value of volume ratio corresponding to different applied strains.

| | applied strain (%) | | | |
|---|---|---|---|---|
| | 5% | 10% | 15% | 20% |
| volume ratio $J = V/V_{ref}$ | 0.981 | 0.925 | 0.860 | 0.760 |

fibre tension parallel to the loading direction. This is due to the large value of the angle between the mean orientation of collagen fibre and the direction of loading (say $\alpha = \sim 83°$). Hence, the large value of fibre tension in the perpendicular direction of loading produces a large lateral expansion against the low-stiffness matrix, which can be responsible for the auxetic behaviour of the tissue. However, in the thickness direction, the deformation behaviour of skin is found similar (lateral contraction) in both parallel and perpendicular direction as in this plane ($Y–Z$ plane) the average value of the angle between collagen fibre mean axis and loading direction is small ($7.5°$ in parallel specimens and $12.5°$ in perpendicular specimens).

The present study also investigates the underlying mechanics of stress relaxation in skin, where the stress relaxation response of pig skin is examined at different strain levels (figure 3a) and corresponding to different loading directions. The stress relaxation behaviour of skin is found to be dependent on the applied strain and loading direction (with respect to STL). In parallel specimens, the amount of relative change in stress is found to be decreasing with an increase in the applied strain. These results show good consistency with the literature [11]. Also, change in tissue volume (decrease in volume of increase in Poisson's ratio ($v_{xy}$ and $v_{yz}$)) during the stress relaxation is observed to be decreasing with an increase in the value of applied strain. The less reduction in volume during the stress relaxation corresponding to the large value of hold strain is the result of less amount of fluid availability in the tissue after a large stretching. As it is clearly observed that the volume of tissue decreases (or loss in IF increases) with an increase in the value of applied strain (table 3). This subsequently results in less amount of available IF in tissue at the end of the loading phase, which leads to less reduction in the volume of tissue (or less loss in tissue fluid) during stress relaxation. These results indicate that the stress dissipation during stress relaxation is due to the loss of IF from the tissue. This statement is confirmed through an *in situ* microscopic study, where the formation of fluid droplets (electronic supplementary material, video-3(a and b)) on the skin surface is noted during the stress relaxation, which is found more corresponding to the small value of holding strain (5%).

Another side, in perpendicular specimens, the large value of stress reduction corresponding to the large value of hold strain (20%) indicates the minimum loss of IF during the stretching. This could be due to the lack of sufficient compressive pressure on the interfibrillar matrix at small and moderate stretching, which is required to push the IF from the tissue. The cause of this small pressure is the large angle of collagen fibres orientation from the loading direction, due to which a large part of applied deformation (approx. 9% strain; this value is corresponding to transition strain) is accumulated in rotation and bending of the fibres, which results in a negligible stretching of interfibrillar cross-linkers and then negligible compressive pressure of the interfibrillar matrix. Also, SEM photographs show delamination of collagen fibres bundles (electronic supplementary material, figure 8S(b)) in tested perpendicular specimens which can also play a role to reduce the amount of compressive pressure on the interfibrillar matrix. However, beyond approximately 9% strain, the value of compressive pressure on the interstitial matrix starts to increase (as beyond the value of this strain, the linear region of stress–strain curve begins and collagen fibres start to stretch), and it can be assumed that most of the part of IF losses during the hold which results in the large value of stress relaxation at the large value of hold strain. This can also be confirmed through the DIC results where the increase in Poisson's ratio (decrease in volume/loss in fluid from the tissue) during the stress relaxation phase is found increasing with the hold strain; also, the increase in Poisson's ratio at 15% and 20% hold strain is significantly larger in perpendicular specimens than parallel specimens.

Further, the relevance of this *ex vivo* study to physiological conditions is confirmed through an *in vivo* study on human mid-back skin. This region is selected because it undergoes large stretching due to the bending of the body during exercise or daily physiological activities. Despite the constraints provided by surrounding tissue to the tested skin, the results of the *in vivo* study show a large lateral contraction/or large Poisson's ratio (>0.5) of skin when loaded in the direction of STL. These results are in good agreement with the *ex vivo* study and confirm the *in vivo* compressibility of the skin. However, like the

*ex vivo* study, the formation of the water droplet on the skin surface is not observed, which can be due to the displacement of fluid from the stretched region to the unstretched surrounding tissue. On the other hand, when skin is stretched perpendicular to STL, the amount of lateral contraction is small, which can be due to the resistance offered by the stiff collagen fibres as they are aligned towards the STL. Furthermore, like the *ex vivo* study, the *in vivo* study does not show the auxetic nature of skin when it is stretched perpendicular to STL. This can be due to the constraints provided by surrounding tissues against rotation as well as elongation of collagen fibres. However, the small value of Poisson's ratio indicates that upon excision, human skin may show auxetic behaviour. Also, during the hold, the *in vivo* study shows an increase in Poisson's ratio (or lateral contraction), which indicates displacement of fluid from the tested region. These observations are found similar in both loading directions; also, the results of the *in vivo* study are found akin to the results of the *ex vivo* study. Therefore, it can be stated that fluid flow is the underlying mechanism of stress relaxation for both *ex vivo* and *in vivo* loading conditions. This *in vivo* study confirms the relevance of the *ex vivo* study results for understanding the mechanical behaviour of human skin during physiological loading or stretching of skin due to surgical interventions.

Despite the new results presented related to the skin tissue, there are numerous limitations associated with this study, First, all the calculations performed in this study are based on the linear elastic theory; however, the skin tissue shows nonlinear elastic response during the stretching [62,63]. The assumption of linear elastic behaviour of skin may have induced some error in the presented results of this study. However, the aim of this study is to investigate the compressible behaviour of skin. Second, we do not considered the effect of loading rate in the calculation of mechanical properties of skin, though skin shows viscoelastic behaviour such as strain rate-dependent material properties. Therefore, the effect of loading rate should also be required to be considered in the calculation of material parameters. Third, during the FEA simulation, the model is considered elastic; however, for capturing of fluid loss from the tissue, the model should be simulated as a biphasic material to predict the fluid loss from the tissue. Therefore, in future, for capturing the more general response of the skin, all these limitations are required to be addressed during the experiments and simulation.

## 5. Conclusion

The results of this study reveal that skin behaves as a compressible material, and its lateral deformation response is direction dependent, which is related to the native configuration of collagen fibres. Furthermore, this study demonstrates that the loss of fluid from the skin is the main driving mechanism of stress relaxation, particularly at small strain levels and the initial stage of stress relaxation. The phenomenon for the later period of stress relaxation, where the increase in Poisson's ratio becomes almost plateau, can be described by the unfolding of collagen fibrils due to redistribution of the applied strain [20]. Also, the results of this study suggest that the skin should be treated as a biphasic material where the collagen fibres, matrix and IF share the induced stresses. This study also suggests that the short stress relaxation during large wound closure can reduce the excessive stresses on wound edges at the expense of fluid displacement/loss. The findings of this study can be helpful in plastic surgery, wound closure, cosmetic industries and skin graft manufacturing.

Data accessibility. The datasets supporting this article have been uploaded as part of the electronic supplementary material [64].

Authors' contributions. K.K.D.: conceptualization, data curation, formal analysis, investigation, methodology, project administration, software, validation, visualization, writing—original draft, writing—review and editing; P.L.: conceptualization, data curation, formal analysis, funding acquisition, investigation, methodology, validation, visualization, writing—review and editing; S.K.: supervision, writing—review and editing; N.K.: resources, supervision, validation, visualization, writing—review and editing.

All authors gave final approval for publication and agreed to be held accountable for the work performed therein.

Competing interests. The authors declare no conflict of interest.

Funding. The research was sponsored by the Ministry of Human Resource Development (MHRD) and the Indian Institute of Technology (IIT) Ropar, India. This study was partially supported by the Department of Higher Education, India (grant no.: STARS1/632).

Acknowledgements. The authors acknowledge highly IIT Ropar for providing the necessary facilities used in this study. The authors would like to acknowledge Mr Piyush Uniyal and Aakash Soni for their help in rectifying the language. The authors also would like to acknowledge Mr Harsimran for performing AFM experiments.

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
