## [Peer Review File · Royal Society Open Science]

Review History

Decision letter (RSOS-210895.R0)

Dear Dr Kumar:

Manuscript ID RSOS-210895 entitled "Effect of collagen fiber orientation on The Poisson's ratio and Stress relaxation of skin: An ex-vivo and in-vivo study" which you submitted to Royal Society Open Science, has been evaluated.

I must decline the manuscript for publication in Royal Society Open Science at this time. The Editor assessing the manuscript would like you to substantially review and revise the language in the manuscript - they are concerned that, at present, the scientific story you are trying to tell is being obscured by unclear and imprecise writing. We would also recommend that you take another look at the structure of the manuscript to help improve its clarity. You may benefit from seeking advice from a professional language editing service such as those available at <https://royalsociety.org/journals/authors/benefits/language-editing/>.

Please note that resubmitting your manuscript does not guarantee eventual acceptance, and that your resubmission will be subject to review by reviewer(s) before a decision is rendered.

You will be unable to make your revisions on the originally submitted version of your manuscript. Instead, revise your manuscript using a word processing program and save it on your computer.

You may also click the below link to start the resubmission process (or continue the process if you have already started your resubmission) for your manuscript. If you use the below link you will not be required to login to ScholarOne Manuscripts.

*** PLEASE NOTE: This is a two-step process. After clicking on the link, you will be directed to a webpage to confirm. ***

https://mc.manuscriptcentral.com/rsos?URL_MASK=1b8d595d49814d92b749d0788bed881d

Because we are trying to facilitate timely publication of manuscripts submitted to Royal Society Open Science, your resubmitted manuscript should be submitted by 21-Dec-2021. If you are unable to submit by this date please contact the Editorial Office for options.

I look forward to a resubmission.

Sincerely,
Royal Society Open Science Editorial Office
openscience@royalsociety.org

Author's Response to Decision Letter for (RSOS-210895.R0)

See Appendix A.

RSOS-211301.R0

Review form: Reviewer 1

Is the manuscript scientifically sound in its present form?

Yes

Are the interpretations and conclusions justified by the results?

Yes

Is the language acceptable?

Yes

Do you have any ethical concerns with this paper?

No

Have you any concerns about statistical analyses in this paper?

No

Recommendation?

Accept with minor revision (please list in comments)

Comments to the Author(s)

Review: Effect of collagen fiber orientation on the Poisson's ratio and stress relaxation of skin: An ex-vivo and in-vivo study

Authors: Krashn Kumar Dwivedi, Piyush Lakhani, Sachin Kumar, and Navin Kumar

This work is a worthy contribution to the ever-growing field of skin biomechanics. The authors presented a rigorous and comprehensive experimental pipeline to explore the skin's viscoelasticity and microstructure using a handful of approaches: uniaxial extension, DIC strain tracking, stress relaxation testing, micro-CT, SEM and AFM, and FEA. I am genuinely impressed with how detailed this study is and with the explanation of many phenomena provided by the authors. For instance, the authors carefully determined the orientation of skin tension lines through an inflation test to have properly oriented uniaxial specimens. In uniaxial tests, the authors tracked not only in-plane deformations but also deformations through thickness. The authors were able to demonstrate the relevance of their ex-vivo animal model results using the in-vivo human case study. The discussion on microstructure and arguments on the volume loss were supported by simulations. The outcome of the study – the observed dependency of stress relaxation response on strain level and directionality; the significance of the fluid loss and microstructure roles in the viscoelastic response. The results and established correlations were scientifically sound, novel and supported by the existing literature. I enjoyed reading the paper, and I recommend the manuscript to be published in the Royal Society Open Science AFTER the authors will address some of my concerns/questions and implement some modifications to improve the quality of the manuscript.

Questions/Concerns:

- Based on uniax experiments videos, the fluid effluxing from skin disturbs the "waterproof" ink speckles, so they detach from the surface of the specimen and start moving with the drops.

How does it impact your strain field calculation?

- Procedure of finding STL: Did you extract 4 circular specimens from the same animal?

Would you expect inter-subject variability or the other pigs would have similar STL?

- Are your uniax sample excised from the same specimens used for the bulge test?

- What do circled numbers in Fig. 1a indicate?

- Your Poisson's ratio and other parameters are based on DIC strains, which are inhomogeneous across the surface. Do you use the averaged strain values in Equations (2-3)?

- Did you track deformations from the epidermis side or the hypodermis side? Based on fig. 2. Y-Z plane, these sides experience different deformations.

Suggestions:

- In general, the discussion section should not have too many new results reported but rather explain results reported in the results section and explore the correlation between them. You are doing a great job following this rule in the last paragraphs of the discussion (stress relaxation, in-vivo, Poisson's ratios). However, the first few paragraphs in the discussion report new results on microstructure, FEA simulation, AFM, fluid drops proof, wrinkled paper analogies. The discussion section starts with "the work explored the correlation of skin microstructure with 3D deformation response and stress relaxation behaviour". However, the ONLY result on the

microstructure reported before this sentence is the collagen orientation. Only after this sentence, you start discussing microstructure. Please move a significant bulk of novel results from the discussion to the results section. You do not have to create subsections for all the results and can combine some of them. Otherwise, you will lose a reader in the discussion part.

- You have a complex methodology (AFM, uniax, bulge, SEM, micro-CT, etc) and maybe the figure-diagram with icons/steps would clarify the experimental pipeline.

Minor notes:

- Fig 4,6 caption contains empty bracket “()”
- Section 4.3. please simplify sentences as it is hard to follow

Review form: Reviewer 2

Is the manuscript scientifically sound in its present form?

Yes

Are the interpretations and conclusions justified by the results?

No

Is the language acceptable?

Yes

Do you have any ethical concerns with this paper?

No

Have you any concerns about statistical analyses in this paper?

No

Recommendation?

Major revision is needed (please make suggestions in comments)

Comments to the Author(s)

The authors perform a comprehensive set of uniaxial tests on porcine skin to determine the change in volume of the skin as it is deformed. This is a relevant topic of investigation since soft tissues are usually considered incompressible which might not be a true assumption. The study is comprehensive and well executed. However, the framework of linear elasticity (Young's modulus, Poisson's ratio calculated in the linear sense, no proper viscoelastic quantification) is too simple for analysis of skin or any soft tissue. Nonetheless, if this is acknowledged in the discussion I still think this is a valuable set of data with some useful insights from the linear analysis. But this has to be addressed as a limitation. The most important contribution, to me, is the change in volume with uniaxial deformation. The data itself is useful too, for other researchers who may want to use more adequate constitutive models and nonlinear elasticity theories to explain the data.

Major comments:

[1] Figure 4d is very difficult to read or understand

[2] Figure 5a. Reduction of stress should be changed to 'relative change in stress' or relative reduction of stress. No units are presented, which is fine if this is a relative metric, but should be clear. 'Reduction in stress' is not clear, as one would expect units of stress (Pa, psi, etc).

[3] Figure 5c,d, which Poisson's ratio?

[4] Figure 5e, caption is the same as for Fig. 5c,d, what is the difference?

[5] Figure 6: it is not clearly presented, is this the same data as Figure 5 just presented in a different way? I don't know if this figure adds to the paper

[6] Figure 9: the finite element model is not really useful since it does not represent a true uniaxial test. For the model to represent a uniaxial test periodic boundary conditions are needed on a representative volume element (which itself must be periodic). Additionally, it is not stated whether this is plane stress or plane strain. Neither seems justifiable, these are stiff intercalated cylindrical fibers in a soft 3D matrix. I'm not sure the current model is representative of the 3D structure. Extreme deformation at the left end of the mesh seems undesirable. Thus, while a good idea in principle, execution in the finite element simulation is far from desirable. I would suggest refining the finite element model or removing completely. Current model is not really representative of the system.

[7] Figure 11 is not very useful. I see the orientation of the fibers relative to the applied load, but this figure does not help me understand any of the observations from the previous figures. What are the authors trying to convey with this figure?

[8] The authors claim, in the discussion, that the FE model explains the reduction in volume. This claim is not correct. To account for volume change due to fluid loss, as claimed, authors need a poro-elastic model. See the work by Dr. Mazza at ETH, they have a couple of papers on poroelastic modeling of skin in which they model volume change in uniaxial test.

[9] Assertion that fibers are oriented along STL based on AFM images is not accurate. The CT image analysis would be a good justification, but AFM is at such a low spatial scale that it is unclear if at this scale one can infer macroscopic anisotropic response. Please see the work by, for example, Dr. Ni Annaidh, and Dr. Nielsen, or Drs. Rausch and Tepole, who have shown how fiber network structure on the order of microns dictate the anisotropy at the macroscale. While I agree that structure observed with AFM is also related to macroscale behavior, I contest the notion that it dictates macroscale anisotropy. It is definitely not conclusively shown.

Minor comments

[1] Skin should not be capitalized in the abstract. There are other capitalized words in the text that do not need to be capitalized.

Decision letter (RSOS-211301.R0)

Dear Dr Kumar

The Editors assigned to your paper RSOS-211301 "Effect of collagen fiber orientation on the Poisson's ratio and stress relaxation of skin: An ex-vivo and in-vivo study" have now received comments from reviewers and would like you to revise the paper in accordance with the reviewer comments and any comments from the Editors. Please note this decision does not guarantee eventual acceptance.

We do not generally allow multiple rounds of revision so we urge you to make every effort to fully address all of the comments at this stage. If deemed necessary by the Editors, your

manuscript will be sent back to one or more of the original reviewers for assessment. If the original reviewers are not available, we may invite new reviewers.

Please submit your revised manuscript and required files (see below) no later than 21 days from today's (ie 23-Nov-2021) date. Note: the ScholarOne system will 'lock' if submission of the revision is attempted 21 or more days after the deadline. If you do not think you will be able to meet this deadline please contact the editorial office immediately.

on behalf of Dr Adil Al-Mayah (Associate Editor) and R. Kerry Rowe (Subject Editor)
openscience@royalsociety.org

Associate Editor Comments to Author (Dr Adil Al-Mayah):

Both reviewers agree on the importance of the conducted study and the value of reported data. However, a number of critical issues need to be addressed. These include a better connection between the original study and the FE models. The discuss and its contents also require the authors' attention as significant new results are added for the first time in the paper without previous mentioning in previous sections.

Reviewer comments to Author:
Reviewer: 1
Comments to the Author(s)

Review: Effect of collagen fiber orientation on the Poisson's ratio and stress relaxation of skin: An ex-vivo and in-vivo study
Authors: Krashn Kumar Dwivedi, Piyush Lakhani, Sachin Kumar, and Navin Kumar

This work is a worthy contribution to the ever-growing field of skin biomechanics. The authors presented a rigorous and comprehensive experimental pipeline to explore the skin's viscoelasticity and microstructure using a handful of approaches: uniaxial extension, DIC strain tracking, stress relaxation testing, micro-CT, SEM and AFM, and FEA. I am genuinely impressed with how detailed this study is and with the explanation of many phenomena provided by the authors. For instance, the authors carefully determined the orientation of skin tension lines through an inflation test to have properly oriented uniaxial specimens. In uniaxial tests, the authors tracked not only in-plane deformations but also deformations through thickness. The authors were able to demonstrate the relevance of their ex-vivo animal model results using the in-vivo human case study. The discussion on microstructure and arguments on the volume loss were supported by simulations. The outcome of the study – the observed dependency of stress

relaxation response on strain level and directionality; the significance of the fluid loss and microstructure roles in the viscoelastic response. The results and established correlations were scientifically sound, novel and supported by the existing literature. I enjoyed reading the paper, and I recommend the manuscript to be published in the Royal Society Open Science AFTER the authors will address some of my concerns/questions and implement some modifications to improve the quality of the manuscript.

Questions/Concerns:

- Based on uniax experiments videos, the fluid effluxing from skin disturbs the “waterproof” ink speckles, so they detach from the surface of the specimen and start moving with the drops. How does it impact your strain field calculation?

- Procedure of finding STL: Did you extract 4 circular specimens from the same animal? Would you expect inter-subject variability or the other pigs would have similar STL?

- Are your uniax sample excised from the same specimens used for the bulge test?

- What do circled numbers in Fig. 1a indicate?

- Your Poisson’s ratio and other parameters are based on DIC strains, which are inhomogeneous across the surface. Do you use the averaged strain values in Equations (2-3)?

- Did you track deformations from the epidermis side or the hypodermis side? Based on fig. 2. Y-Z plane, these sides experience different deformations.

Suggestions:

- In general, the discussion section should not have too many new results reported but rather explain results reported in the results section and explore the correlation between them. You are doing a great job following this rule in the last paragraphs of the discussion (stress relaxation, in-vivo, Poisson’s ratios). However, the first few paragraphs in the discussion report new results on microstructure, FEA simulation, AFM, fluid drops proof, wrinkled paper analogies. The discussion section starts with “the work explored the correlation of skin microstructure with 3D deformation response and stress relaxation behaviour”. However, the ONLY result on the microstructure reported before this sentence is the collagen orientation. Only after this sentence, you start discussing microstructure. Please move a significant bulk of novel results from the discussion to the results section. You do not have to create subsections for all the results and can combine some of them. Otherwise, you will lose a reader in the discussion part.

- You have a complex methodology (AFM, uniax, bulge, SEM, micro-CT, etc) and maybe the figure-diagram with icons/steps would clarify the experimental pipeline.

Minor notes:

- Fig 4,6 caption contains empty bracket “()”
- Section 4.3. please simplify sentences as it is hard to follow

Reviewer: 2

Comments to the Author(s)

The authors perform a comprehensive set of uniaxial tests on porcine skin to determine the change in volume of the skin as it is deformed. This is a relevant topic of investigation since soft tissues are usually considered incompressible which might not be a true assumption. The study is comprehensive and well executed. However, the framework of linear elasticity (Young’s modulus, Poisson’s ratio calculated in the linear sense, no proper viscoelastic quantification) is too simple for analysis of skin or any soft tissue. Nonetheless, if this is acknowledged in the discussion I still think this is a valuable set of data with some useful insights from the linear analysis. But this has to be addressed as a limitation. The most important contribution, to me, is the change in volume with uniaxial deformation. The data itself is useful too, for other researchers who may want to use more adequate constitutive models and nonlinear elasticity theories to explain the data.

Major comments:

- [1] Figure 4d is very difficult to read or understand
- [2] Figure 5a. Reduction of stress should be changed to 'relative change in stress' or relative reduction of stress. No units are presented, which is fine if this is a relative metric, but should be clear. 'Reduction in stress' is not clear, as one would expect units of stress (Pa, psi, etc).
- [3] Figure 5c,d, which Poisson's ratio?
- [4] Figure 5e, caption is the same as for Fig. 5c,d, what is the difference?
- [5] Figure 6: it is not clearly presented, is this the same data as Figure 5 just presented in a different way? I don't know if this figure adds to the paper
- [6] Figure 9: the finite element model is not really useful since it does not represent a true uniaxial test. For the model to represent a uniaxial test periodic boundary conditions are needed on a representative volume element (which itself must be periodic). Additionally, it is not stated whether this is plane stress or plane strain. Neither seems justifiable, these are stiff intercalated cylindrical fibers in a soft 3D matrix. I'm not sure the current model is representative of the 3D structure. Extreme deformation at the left end of the mesh seems undesirable. Thus, while a good idea in principle, execution in the finite element simulation is far from desirable. I would suggest refining the finite element model or removing completely. Current model is not really representative of the system.
- [7] Figure 11 is not very useful. I see the orientation of the fibers relative to the applied load, but this figure does not help me understand any of the observations from the previous figures. What are the authors trying to convey with this figure?
- [8] The authors claim, in the discussion, that the FE model explains the reduction in volume. This claim is not correct. To account for volume change due to fluid loss, as claimed, authors need a poro-elastic model. See the work by Dr. Mazza at ETH, they have a couple of papers on poroelastic modeling of skin in which they model volume change in uniaxial test.
- [9] Assertion that fibers are oriented along STL based on AFM images is not accurate. The CT image analysis would be a good justification, but AFM is at such a low spatial scale that it is unclear if at this scale one can infer macroscopic anisotropic response. Please see the work by, for example, Dr. Ni Annaidh, and Dr. Nielsen, or Drs. Rausch and Tepole, who have shown how fiber network structure on the order of microns dictate the anisotropy at the macroscale. While I agree that structure observed with AFM is also related to macroscale behavior, I contest the notion that it dictates macroscale anisotropy. It is definitely not conclusively shown.

Minor comments

- [1] Skin should not be capitalized in the abstract. There are other capitalized words in the text that do not need to be capitalized.

===PREPARING YOUR MANUSCRIPT===

If you have been asked to revise the written English in your submission as a condition of publication, you must do so, and you are expected to provide evidence that you have received language editing support. The journal would prefer that you use a professional language editing service and provide a certificate of editing, but a signed letter from a colleague who is a fluent speaker of English is acceptable. Note the journal has arranged a number of discounts for authors using professional language editing services (<https://royalsociety.org/journals/authors/benefits/language-editing/>).

===PREPARING YOUR REVISION IN SCHOLARONE===

- Ensure that your data access statement meets the requirements at <https://royalsociety.org/journals/authors/author-guidelines/#data>. You should ensure that you cite the dataset in your reference list. If you have deposited data etc in the Dryad repository, please include both the 'For publication' link and 'For review' link at this stage.
- If you are requesting an article processing charge waiver, you must select the relevant waiver option (if requesting a discretionary waiver, the form should have been uploaded at Step 3 'File upload' above).
- If you have uploaded ESM files, please ensure you follow the guidance at <https://royalsociety.org/journals/authors/author-guidelines/#supplementary-material> to include a suitable title and informative caption. An example of appropriate titling and captioning may be found at https://figshare.com/articles/Table_S2_from_Is_there_a_trade-off_between_peak_performance_and_performance_breadth_across_temperatures_for_aerobic_sc_ope_in_teleost_fishes_/3843624.

Author's Response to Decision Letter for (RSOS-211301.R0)

See Appendix B.

RSOS-211301.R1

Review form: Reviewer 1

Is the manuscript scientifically sound in its present form?

Yes

Are the interpretations and conclusions justified by the results?

Yes

Is the language acceptable?

Yes

Do you have any ethical concerns with this paper?

No

Have you any concerns about statistical analyses in this paper?

No

Recommendation?

Accept as is

Comments to the Author(s)

Thank you for thoroughly addressing my comments. I recommend the manuscript for acceptance

Review form: Reviewer 2

Is the manuscript scientifically sound in its present form?

Yes

Are the interpretations and conclusions justified by the results?

Yes

Is the language acceptable?

Yes

Do you have any ethical concerns with this paper?

No

Have you any concerns about statistical analyses in this paper?

No

Recommendation?

Accept as is

Comments to the Author(s)

The authors addressed my concerns

Decision letter (RSOS-211301.R1)

Dear Dr Kumar,

It is a pleasure to accept your manuscript entitled "Effect of collagen fiber orientation on the Poisson's ratio and stress relaxation of skin: An ex-vivo and in-vivo study" in its current form for publication in Royal Society Open Science. The comments of the reviewer(s) who reviewed your manuscript are included at the foot of this letter.

on behalf of Dr Adil Al-Mayah (Associate Editor) and R. Kerry Rowe (Subject Editor)
openscience@royalsociety.org

Reviewer comments to Author:

Reviewer: 1

Comments to the Author(s)

Thank you for thoroughly addressing my comments. I recommend the manuscript for acceptance

Reviewer: 2

Comments to the Author(s)

The authors addressed my concerns

Appendix A

Letter of response to the editor

RSOS-211301 - Effect of collagen fiber orientation on the Poisson's ratio and stress relaxation of skin: An ex-vivo and in-vivo study

The editor comments

We had received the editor comments only, where the revision of language were suggested. The editor comments are as

Manuscript ID RSOS-210895 entitled "Effect of collagen fiber orientation on The Poisson's ratio and Stress relaxation of skin: An ex-vivo and in-vivo study" which you submitted to Royal Society Open Science, has been evaluated.

I must decline the manuscript for publication in Royal Society Open Science at this time. The Editor assessing the manuscript would like you to substantially review and revise the language in the manuscript - they are concerned that, at present, the scientific story you are trying to tell is being obscured by unclear and imprecise writing. We would also recommend that you take another look at the structure of the manuscript to help improve its clarity. You may benefit from seeking advice from a professional language editing service such as those available at <https://royalsociety.org/journals/authors/benefits/language-diting/>.

Please note that resubmitting your manuscript does not guarantee eventual acceptance, and that your resubmission will be subject to review by reviewer(s) before a decision is rendered.

You will be unable to make your revisions on the originally submitted version of your manuscript. Instead, revise your manuscript using a word processing program and save it on your computer.

You may also click the below link to start the resubmission process (or continue the process if you have already started your resubmission) for your manuscript. If you use the below link you will not be required to login to ScholarOne Manuscripts.

**** PLEASE NOTE: This is a two-step process. After clicking on the link, you will be directed to a webpage to confirm. *****

Response to the editor

Authors are thankful to the editor to reconsider the manuscript for the publication in the journal. Authors had carefully address the editor suggestions. In responses, the line numbers are corresponding to the resubmitted manuscript.

The modification in the resubmitted manuscript are highlighted along with marked comments. At the time of revision we did not activate the change tracker in MS word, therefore, the changes in the resubmitted manuscript corresponding to old version have been highlighted. The section wise modification in resubmitted manuscripts are;

1. Summary

As per the editor suggestions related to language, whole paragraph (**line no 11-24**) has been rewritten in resubmitted manuscript to improve the language.

2. Introduction

Line no 27-31 sentences have been modified for better understanding and improving the language.

Line no 35-36 sentence has been rephrase to fix the grammatical error.

Line no 40-42 sentence has been modified for conveying the clear message.

Line no 43 wording has been changed for making full sense.

Line no 44 word has been modified.

Line no 58, 60, and 64 word has been rephrased.

Line no 68-70 sentences have been rephrased to fix grammatical error.

Line no 71 word has been rephrased.

Line no 74-84 sentences have been modified and rephrased for more clear meaning and to fix the grammatical issues.

3. Materials and methods

3.1 *Sample preparation*, **Line no 88-109** whole paragraph has been rephrased to fix grammatical error and to improve the language.

3.2 *Ex-vivo experimental setup* **Line no 111** title of the section has been changed from 'experimental setup' to 'Ex-vivo experimental setup'.

Line no 113 words have been added to make more clear sense.

Line no 114-123 sentences have been rephrased to improve the language.

3.3 *Digital Image correlation (DIC)*, this section has been sifted from \$ 3.5 to 3.3 for making the continuity.

Line no 126-127 words (parallel to skin surface, and plane of thickness measurement) have been added to make more clear sense.

Line no 137-140 paragraph has been rewritten for making clear description.

3.4 *Ex-vivo loading and stress relaxation*, **Line no 146-147, 149-153**, sentence have been modified.

3.5 *In-vivo experiment on human skin*, numbering of the section has been changed from 3.4 to 3.5.

Line no 155-158 sentences have been rewritten to make clear sense.

Line no 160-162 sentence has been modify to fix grammatical errors as per editor suggestions.

Line no 165-166, 167-168 sentence has been modified for the clear interpretation.

3.6 *Micro-CT and image processing*, **Line no 171-172** sentence has been rephrased.

Line no 178, 180 (staining, which was) word has been added.

Line no 180, 182 sentence has been modified.

3.8 *Modelling of the dermis deformation response at collagen fibrils scale*

Line no 204-205 sentence has been modified.

Line no 208 word has been rephrased.

3.9 *Data analysis*

Line no 219 word has been rephrased.

Line no 220-222 sentence has been modified.

Line no 223 word has been added.

Line no 228- 229 sentence has been modified.

Line no 232 words (during the hold period) have been added.

3.10 *Statistical analysis*

Line no 234 article 'the' has been added,

Line no 236, 244 words have been rephrased.

4 Results

4.1 *Loading portion*

Line no 249-252 sentences have been modified.

Line no 254-255 sentences have been modified.

Line no 256,258 words have been rephrase.

Line no 260-264 sentences have been modified.

Line no 267-277 sentences have been modified.

Line no 280-281 sentences have been added.

Line no 282 word has been rephrased.

4.2 Stress relaxation portion

Line no 286 initial latter of words have been rewritten in lower case.

Line no 291-293 sentence has been modified.

Line no 294 word has been rephrased.

Line no 295 sentence has been rephrased.

Line no 296 words has been rephrased.

Line no 297 sentence has been rephrased.

Line no 298 word 'on the other hand' has been added.

Line no 310-312 sentence has been modified.

Line no 320-322 sentence has been modified to fix grammatical error.

4.3 Results of in vivo experiment.

Paragraph of this been rewritten for improving the language quality.

4.4 Collagen orientation

Line no 343-348 sentences have been modified to fix grammatical error.

5 Discussion

Please note that whole discussion section has been converted in present form.

Line no 351-355 sentence has been modified.

Line no 356-360 sentence has been modified and typo error has been fixed by replacing '75% to 25%'.

Line no 361 sentence has been modified.

Line no 364 word 'be' has been added.

Line no 371 Sentence has been modified and merged with previous sentence.

Line no 373 Sentence has been modified by replacing the word 'pressure gradient' with pressure and by removing words 'with respect to its environment' as these words are misleading the exact meaning of sentences.

Line no 378-383 Sentences have been modified to eliminate the grammatical error.

Line no 386-454 whole paragraphs have been modified to improve the language.

Line no 456-463 Sentence has been rephrased and modified.

Line no 464 word 'Furthermore' has been added.

Line no 466, 467 word has been rephrased.

Line no 468, 469 sentence have been rephrased.

Line no 470 word has been rephrased.

Line no 472-474 sentences have been modified and rewritten.

6 Conclusion

Whole conclusion section has been rewritten to improve the language.

Figure captions

Line no 1088-1090 (Fig.5) words have been rephrased.

Line no 1096-1098 (Fig 8) sentences have been rephrased.

Line no 1099-1104 (Fig 9) sentences have been rephrased.

Appendix B

Summary of Revisions

Manuscript No: RSOS-211301

**Title: Effect of collagen fiber orientation on the Poisson's ratio and stress relaxation of skin:
An ex-vivo and in-vivo study**

We are thankful to the editor and reviewers for their fruitful comments and recommendations. The authors have carefully addressed the reviewer's suggestions and comments. We strongly feel that the manuscript has been improved by addressing the revisions. We addressed the reviewers' specific concerns in tabular format. Modifications in the revised manuscript are highlighted in yellow colour. Page and line numbers in the explanation are corresponding to the revised manuscript.

Reviewer # 1

This work is a worthy contribution to the ever-growing field of skin biomechanics. The authors presented a rigorous and comprehensive experimental pipeline to explore the skin's viscoelasticity and microstructure using a handful of approaches: uniaxial extension, DIC strain tracking, stress relaxation testing, micro-CT, SEM and AFM, and FEA. I am genuinely impressed with how detailed this study is and with the explanation of many phenomena provided by the authors. For instance, the authors carefully determined the orientation of skin tension lines through an inflation test to have properly oriented uniaxial specimens. In uniaxial tests, the authors tracked not only in-plane deformations but also deformations through thickness. The authors were able to demonstrate the relevance of their ex-vivo animal model results using the in-vivo human case study. The discussion on microstructure and arguments on the volume loss were supported by simulations. The outcome of the study – the observed dependency of stress relaxation response on strain level and directionality; the significance of the fluid loss and microstructure roles in the viscoelastic response. The results and established correlations were scientifically sound, novel and supported by the existing literature. I enjoyed reading the paper, and I recommend the manuscript to be published in the Royal Society Open Science AFTER the authors will address some of my concerns/questions and implement some modifications to improve the quality of the manuscript.

The authors are thankful to the reviewer for encouraging comments and suggestions. The authors have incorporated all suggestions and changes in the revised manuscript. We feel that after incorporation the suggestions and changes recommended by the reviewer, the quality of the manuscript has been improved.

S.No.	COMMENT	EXPLANATION
1	GENERAL COMMENTS: Based on uniax experiments videos, the fluid effluxing from skin disturbs the “waterproof” ink speckles, so they detach from the surface of the specimen and start moving with the drops. How does it impact your strain field calculation?	The authors are thankful to the reviewer for pointing out the missing explanation from the manuscript related to DIC and microscopic experiments. We performed two separate imaging tests: 1) DIC test to measure the strains and 2) microscopic test to observe the fluid exudation during the ramp and relaxation.  1. DIC: for measuring the strains through the DIC, random black dots were made using waterproof ink. These dots were applied to the epidermis side of the skin. During stretching of the specimens, some small bubbles were observed on the skin surface, although no detachment of dots was observed during the stretching. Therefore, to confirm the effect of fluid bubbles on strain calculation, we performed a supplementary test on the metal plate. The black dots similar (e.g. random size and space) to the skin specimens were made on the plate surface and then the plate was fixed vertically on the lower grip of the tensile test machine. Twenty to thirty images of the plate were captured and subsequently, water was sprayed on the surface of the plate (to mimic the formation of fluid bubble on the skin surface) and again twenty

to thirty images of water moistened surface was captured. Using DIC, the captured images were analyzed and the strain was measured for both the conditions (with and without water). The measured values of strain in dry and wet cases were 0.001 and 0.0019, respectively. These values of strain can be considered almost equal to zero which is expected as in both cases plate was not deformed. This small error in strain due to water/fluid flow can be neglected during the finite deformation of specimens.

Also, at the some points of the specimen, a disturbance in correlation was observed due to the formation of water droplets but it did not affect the average value of strain.

2. Microscopic: for capturing the fluid exudation during ramp and relaxation, the specimens (n=5, parallel to STL and n=4, perpendicular to STL) adjacent to the stress relaxation test specimens were excised and black paint was sprayed on the skin surface of the epidermis side. Skin specimens were stretched up to different strain values and held at these values of strains. The fluid exudation during the stretch and hold was captured by replacement and movement of paint as observed in video 1 and video 3 (a&b). These specimens were not included in strain calculation and this experiment only provides evidence of fluid exudation during ramp and hold. This missing

		explanation has been added in revised manuscript (Line no 140-164).
2.	Procedure of finding STL: Did you extract 4 circular specimens from the same animal? Would you expect inter-subject variability or the other pigs would have similar STL?	The four specimens for the bulge test were excised from the two different animals (two specimens from each animal) of the same age and similar to uniaxial tests specimens, these four specimens were also extracted from the dorsal portion of animals. The authors agree with the reviewer concern about the inter-animal variation in STLs directions. We observed some variation in the direction of STLs between the animals though this variation was not found significant. However, for presenting the accurate effect of STLs orientation on the calculated mechanical properties of skin, the direction of STLs was not generalized between the animals and marked in each animal individually. The missing information is updated in the revised manuscript (Line no 79, 87-90 and 94-97).
3.	Are your uniax sample excised from the same specimens used for the bulge test?	Yes, we have excised the uniaxial test specimens from the same animals' skin from where the specimens for the bulge test were excised. The missing information is added in the revised manuscript (Line no 87-90).
4.	What do circled numbers in Fig. 1a indicate?	The authors are grateful to the reviewer for pointing out the missing information. Circle number 1-5 in blue circles represent the random locations of five specimens that were excised parallel to STLs whereas circle 1-4 in green circles represents the random locations of four specimens, were excised

		perpendicular to STLs. These specimens were used for microstructural assessment of skin This information has been incorporated in the revised manuscript (caption of Fig 1(a)).
5	Your Poisson's ratio and other parameters are based on DIC strains, which are inhomogeneous across the surface. Do you use the averaged strain values in Equations (2-3)?	The authors are thankful to the reviewer for pointing the missing details. Yes, Eq (2-3) used the average of all localized strains across the surface. The localized values of strain were averaged over the chosen area of interest. This area was selected far from the boundaries of the specimens, which eliminates the boundary effect in strain calculation. The Area of interest over which the strain was averaged has been marked in Fig 3 of revised manuscript. The missing information has been incorporated in the main manuscript (Line no 264-267,270-271).
6	Did you track deformations from the epidermis side or the hypodermis side? Based on fig. 2. Y-Z plane, these sides experience different deformations.	The deformation of the X-Y plane was tracked from the epidermis side only. As per reviewer suggestions we measured the deformation of epidermis and hypodermis sides in the Y-Z plane along the lines as shown in Figure 1 (A). After averaging the deformation over lines, we found an almost similar value of deformation on both sides. The difference in the distribution of local strains between the epidermis side and the hypodermis side may be due to the boundary effect. During strain calculation, the Area of interest over which strain was averaged was taken away

from the boundaries. Please see Figure 1(B).

Figure 1 (A) average deformation measured epidermis and hypodermis side along the line (B) selected area of interest (black box) over which the average value of strain was measured.

This information has been incorporated in revised manuscript in Line no 126.

1 SUGGESTIONS:

In general, the discussion section should not have too many new results reported but rather explain results reported in the results section and explore the correlation between them. You are doing a great job following this rule in the last paragraphs of the discussion (stress relaxation, in-vivo, Poisson's ratios). However, the first few paragraphs in the discussion report new results on microstructure, FEA simulation, AFM, fluid drops proof, wrinkled

The authors are thankful to the reviewer for this fruitful suggestion. Apart from the discussion, we have incorporated the results related to AFM, SEM, Fluid exudation, wrinkle paper analogy and FEA in the result section.

The result related to collagen fibre and fibril arrangement obtained from SEM and AFM, respectively have been added in section 4.4 under the title of "collagen fibre orientation, collagen fibre and fibril arrangement". The observations related to the white paper wrinkle analogy was also added in this section (Line no 396-404)

	paper analogies. The discussion section starts with “the work explored the correlation of skin microstructure with 3D deformation response and stress relaxation behaviour”. However, the ONLY result on the microstructure reported before this sentence is the collagen orientation. Only after this sentence, you start discussing microstructure. Please move a significant bulk of novel results from the discussion to the results section. You do not have to create subsections for all the results and can combine some of them. Otherwise, you will lose a reader in the discussion part.	The results related to fluid efflux has been added in the section 4.5 under the title of “Exudation of fluid from the specimen”. (Line no 406-411). The FEA results have been incorporated in section 4.6 under the title of “FEA simulation” (Line no 413-428).
2	You have a complex methodology (AFM, uniax, bulge, SEM, micro-CT, etc) and maybe the figure-diagram with icons/steps would clarify the experimental pipeline	The authors are grateful to the reviewer for suggesting the incorporation of a pipeline of experiments (Figure 2). This schematic representation of the experimental pipeline has been incorporated in the revised manuscript in Fig. 2.

Figure 2 Representation of experimental pipeline

1	MINOR NOTES Fig 4,6 caption contains empty bracket “()”	We added the missing contains inside the bracket in both Fig. 4 and Fig. 6. Please note. In the revised manuscript, Fig 4 becomes Fig 5 and Fig 6 has been removed and data related to this figure have been mentioned in the test of the result section.
2	Section 4.3. please simplify sentences as it is hard to follow	As per the reviewer suggestion, the sentences in section 4.3 have been rephrased to make a clear understanding. (Line no 374-386).

Reviewer #2

The authors perform a comprehensive set of uniaxial tests on porcine skin to determine the change in volume of the skin as it is deformed. This is a relevant topic of investigation since soft tissues are usually considered incompressible which might not be a true assumption. The study is comprehensive and well

executed. However, the framework of linear elasticity (Young's modulus, Poisson's ratio calculated in the linear sense, no proper viscoelastic quantification) is too simple for analysis of skin or any soft tissue. Nonetheless, if this is acknowledged in the discussion I still think this is a valuable set of data with some useful insights from the linear analysis. But this has to be addressed as a limitation. The most important contribution, to me, is the change in volume with uniaxial deformation. The data itself is useful too, for other researchers who may want to use more adequate constitutive models and nonlinear elasticity theories to explain the data.

The authors are grateful to the reviewer for encouraging comments and suggestions. The limitations of this study have been incorporated in the last paragraph of the revised manuscript (Line no 557-568).

Major concerns

1.	Figure 4d is very difficult to read or understand	As per reviewer suggestions, Fig 5 (d) has been bifurcated in two separate figures (Fig 5 (d) and Fig.5 (e)) and the numbering of other figures in this panel is also updated accordingly. Please note , in the revised manuscript, Fig. 4 becomes Fig. 5
2.	Figure 5a. Reduction of stress should be changed to 'relative change in stress' or relative reduction of stress. No units are presented, which is fine if this is a relative metric, but should be clear. 'Reduction in stress' is not clear, as one would expect units of stress (Pa, psi, etc).	The authors are thankful to the reviewer for pointing out the confusing matter. As per the reviewer suggestion, the Reduction of stress has been replaced by the relative change in stress in Fig 6 a. This word is also replaced throughout the manuscript. Please note , in the revised manuscript, Fig. 5 becomes Fig. 6.
3.	Figure 5c,d, which Poisson's ratio?	We have incorporated the missing information about Poisson's ratio in Fig 6 (c and d). Fig 6 (c) presents the increase in Poisson's ratio (ν_{xy}) for specimens parallel to STLs whereas Fig 6 (d) presents an increase in Poisson's ratio (ν_{xy}) for specimens perpendicular to STLs.

		The same information has also been updated in the caption of Fig 6. Please note, in the revised manuscript, Fig. 5 becomes Fig. 6.
4.	Figure 5e, caption is the same as for Fig. 5c,d, what is the difference?	The authors are grateful to the reviewer for pointing out the missing information. Fig. 6 (c&d) present the results of Increase in Poisson’s ratio (v_{xy}) in parallel and perpendicular specimens whereas Fig 6 (e) presents results of an Increase in Poisson’s ratio (v_{yz}) in parallel and perpendicular specimens. The captions of these figures have been updated accordingly in the revised manuscript.
5.	Figure 6: it is not clearly presented, is this the same data as Figure 5 just presented in a different way? I don’t know if this figure adds to the paper.	This figure represents the comparison of Relative change in stress and Increase in Poisson’s ratio between parallel and perpendicular specimens. Yes, Fig. 6 (according to old version of manuscript) presents the same data as presented in Fig. 5 (according to old version of manuscript). Therefore, as per the reviewer suggestion, Fig 6 (according to old version of manuscript) has been removed from the manuscript and required information e.g. p-value (for presenting the statistical difference in measured parameters between parallel and perpendicular specimens) related to this figure has been added in results sections 4.2 of the updated manuscript (Line no 360-368).
6	Figure 9: the finite element model is not really useful since it does not represent a true uniaxial test. For the model to represent a uniaxial test periodic boundary conditions are needed on a representative volume	The authors are grateful to the reviewer for providing an accurate direction to model the collagen fibrils and matrix volumetric representative. As per the reviewer suggestions, 3D cylindrical fibrils are assembled within a soft 3D matrix in a staggered pattern

element (which itself must be periodic). Additionally, it is not stated whether this is plane stress or plane strain. Neither seems justifiable, these are stiff intercalated cylindrical fibers in a soft 3D matrix. I'm not sure the current model is representative of the 3D structure. Extreme deformation at the left end of the mesh seems undesirable. Thus, while a good idea in principle, execution in the finite element simulation is far from desirable. I would suggest refining the finite element model or removing completely. Current model is not really representative of the system.

(Figure-3). The collagen fibrils are interconnected through spring elements (representation of cross linker e.g. Glycosaminoglycans (GAGs)). Two left side collagen fibrils (marked in Figure-3(a)) are kept unconnected over some length. This is done purposely to observe the role of cross linkers in collagen fibrils deformation (as we thought that under the tensile force, the cross linkers pull the neighboring collagen fibrils relative to each other). The sliding between the collagen fibrils and matrix is allowed by providing finite sliding-type interaction between matrix and fibrils. The penetration of fibrils on soft matrix during the deformation is avoided by defining a hard contact between fibrils and matrix surfaces.

This model has symmetric surfaces perpendicular to all three Cartesian coordinates. Model is considered elastic, where the material properties of matrix, fibrils and cross linker are given as

Matrix material properties- Young's modulus = 0.14 MPa, Poisson's ratio 0.3.

Fibril material properties- Young's modulus = 1.5 GPa, Poisson's ratio 0.3

Cross linker properties (model as spring element)- Stiffness = 1.3×10^{-3} N/m.

These material properties are adopted from the literature [1,2-3]

For incorporating the infinite size body behaviour during the deformation, the periodic boundary condition was applied. A method similar to the work Pahr and Zysset (2008) [4] was used to apply the periodic boundary

conditions. The model is meshed using the tetrahedron element with a mesh size of 0.5 μm (Supplementary Fig 7S) and simulated under plane strain conditions. A tensile strain of 0.06 was applied along the Z-axis.

The obtained results showed that under deformation all collagen fibrils (except two unconnected left side fibrils) come closer to each other, which is due to the stretching of cross linkers under the tension (see, Figure 3 and supplementary Fig. 7S). The center distance between the collagen fibrils is found to decrease with increase in applied strain. The relative pulling of collagen fibrils squeezes the encapsulated matrix. The deformed 3D view of RVE is provided in supplementary Fig 7S (c-d).

The detail about the FEA simulation has been incorporated in revised manuscript (Line no 229-251, 413-428 and 440-459). This figure (Figure 3) has been incorporated in revised manuscript as Fig. 11.

Figure 3 (a) The volumetric representative (RVE) model of collagen fibril and matrix assembly. The collagen fibrils assembled in parallel and staggered manner inside the soft matrix. The RVE with tetrahedron mesh is presented in supplementary Fig 7S (a). (b) A cross-sectional view RVE at undeformed state. Cross-sectional view of RVE at deformed states corresponding to applied strain (c) 0.02 (d) 0.04 (e) 0.06. The 3-D view of RVE in deformed state is provided in supplementary Fig 7S (c-d). The all neighboring collagen fibrils except unconnected fibrils (two left side) come closure to each other under the stretching of RVE.

7. Figure 11 is not very useful. I see the orientation of the fibers relative to the applied load, but this figure does not help me understand any of the observations from the previous

The schematic in Fig 11 (corresponding to old version of manuscript) was used to explain the different mechanics of collagen fibres and matrix deformation when the skin was loaded parallel and perpendicular to skin tension lines. As per the reviewer concern, this figure has been

	figures. What are the authors trying to convey with this figure?	removed from the main manuscript and related text in manuscript has been modified accordingly.
8.	The authors claim, in the discussion, that the FE model explains the reduction in volume. This claim is not correct. To account for volume change due to fluid loss, as claimed, authors need a poro-elastic model. See the work by Dr. Mazza at ETH, they have a couple of papers on Poroelastic modeling of skin in which they model volume change in uniaxial test.	The authors are thankful to the reviewer for pointing out the over-claimed statements. As per the reviewer suggestion, we read the work of Dr Mazza at ETH related to Poroelastic modelling of skin and other soft tissues. In this study, we did not use Poroelastic theory during the FEA simulation, which is one of the limitations of this study. However, the aim of the FEA simulation is to observe the deformation response of collagen fibrils due to tensioning of cross linker under the tensile loading and to relate it with the cause of fluid exudation from the tissue. This model does not predict the volume loss or fluid loss of/from the tissue and over-optimistic sentences have been rephrased in revised manuscript (Line no 440-459).
9.	Assertion that fibers are oriented along STL based on AFM images is not accurate. The CT image analysis would be a good justification, but AFM is at such a low spatial scale that it is unclear if at this scale one can infer macroscopic anisotropic response. Please see the work by, for example, Dr. Ni Annaidh, and Dr. Nielsen, or Drs. Rausch and Tepole, who have shown how fiber network structure on the order of microns dictate the anisotropy at the macroscale. While I agree that structure observed with AFM is also	As per the reviewer suggestion, the authors have thoroughly read the suggested literature. The authors agree with the reviewer concerns that the orientation of collagen fibrils is not an accurate measurement of STLs orientation or anisotropy. In this study, AFM was used to capture the arrangement of collagen fibrils within the fibres. Most collagen fibrils were found parallelly arranged in fibres. Based on this information, we developed a finite element model, where the collagen fibrils were kept in a parallel arrangement. AFM obtained results were not used to measure the anisotropy of skin tissue. The misleading statements in the

related to macroscale behavior, I contest the notion that it dictates macroscale anisotropy. It is definitely not conclusively shown	updated manuscript have been rephrased (Line no.443-444).
--	---

Minor Concerns

Skin should not be capitalized in the abstract. There are other capitalized words in the text that do not need to be capitalized.	The authors are thankful to the reviewer for pointing out the typo error. As the reviewer suggests, the whole manuscript has been reviewed carefully and all typo errors have been fixed in the updated manuscript.
---	---

Refernces

1. Aziz J, Ahmad MF, Rahman MT, Yahya NA, Czernuszka J, Radzi Z. 2018 AFM analysis of collagen fibrils in expanded scalp tissue after anisotropic tissue expansion. *Int. J. Biol. Macromol.* **107**, 1030–1038.
2. Ault HK, Hoffman AH. 1992 A composite micromechanical model for connective tissues: part II—application to rat tail tendon and joint capsule.
3. Redaelli A, Vesentini S, Soncini M, Vena P, Mantero S, Montevecchi FM. 2003 Possible role of decorin glycosaminoglycans in fibril to fibril force transfer in relative mature tendons—a computational study from molecular to microstructural level. *J. Biomech.* **36**, 1555–1569.
4. Pahr DH, Zysset PK. Influence of boundary conditions on computed apparent elastic properties of cancellous bone. *Biomech Model Mechanobiol.* 2008 Dec;7(6):463-76. doi: 10.1007/s10237-007-0109-7. Epub 2007 Oct 31. PMID: 17972122.